# Differential functions of FANCI and FANCD2 ubiquitination stabilize ID2 complex on DNA

Martin L Rennie[1], Kimon Lemonidis[1], Connor Arkinson[1], Viduth K Chaugule[1], Mairi Clarke[2], James Streetley[2] (iD), Laura Spagnolo[1] & Helen Walden[1,*] (iD)

## Abstract

The Fanconi anaemia (FA) pathway is a dedicated pathway for the repair of DNA interstrand crosslinks and is additionally activated in response to other forms of replication stress. A key step in the FA pathway is the monoubiquitination of each of the two subunits (FANCI and FANCD2) of the ID2 complex on specific lysine residues. However, the molecular function of these modifications has been unknown for nearly two decades. Here, we find that ubiquitination of FANCD2 acts to increase ID2's affinity for double-stranded DNA via promoting a large-scale conformational change in the complex. The resulting complex encircles DNA, by forming a secondary "Arm" ID2 interface. Ubiquitination of FANCI, on the other hand, largely protects the ubiquitin on FANCD2 from USP1-UAF1 deubiquitination, with key hydrophobic residues of FANCI's ubiquitin being important for this protection. In effect, both of these post-translational modifications function to stabilize a conformation in which the ID2 complex encircles DNA.

**Keywords** deubiquitination; DNA repair; Fanconi anaemia; protein–DNA interactions; ubiquitin

**Subject Categories** DNA Replication, Recombination & Repair; Post-translational Modifications & Proteolysis; Structural Biology

## Introduction

Repair of DNA damage is an important aspect of cellular biology, and numerous pathways have evolved to combat different types of DNA damage [1]. Fanconi anaemia (FA) is a rare genetic disorder that arises due to mutations within any of the Fanconi anaemia complementation group (FANC) genes, the products of which are involved in repair of DNA interstand crosslinks (ICLs) [2,3] as well as in the maintenance of genomic stability in response to replication stress [4,5]. While quite rare in the general population, FA pathway genes are frequently altered in cancer patients [3].

A key step in this pathway is the ubiquitination of a pair of paralogous proteins, FANCI (~ 150 kDa) and FANCD2 (~160 kDa) [6,7], which promotes their retention on chromatin [7,8]. In particular, ubiquitination of FANCD2 has been shown to be indispensable for cellular resistance to mitomycin C [6,9], which promotes ICLs. Unlike typical ubiquitination events, FANCI and FANCD2 are each specifically monoubiquitinated at a single conserved lysine. Ube2T-FANCL are the E2–E3 pair that mediate ubiquitination [10,11]. In many eukaryotes, including humans, FANCL is incorporated into a pseudo-dimeric ~ 1 MDa complex which is known as the FA core complex [12,13, preprint: 14]. Removal of the ubiquitins, on FANCI and FANCD2, is also critical for the FA pathway, and this deubiquitination step is catalysed by the USP1-UAF1 complex [15,16].

Evidence suggests FANCI and FANCD2 are involved in recruitment of other proteins [8,17]. However, the mechanistic and structural details of the role of ubiquitination remain ambiguous. The two proteins have been shown to associate *in vivo* [7] and form a heterodimer *in vitro* [18]. A crystal structure of the non-ubiquitinated mouse FANCI-FANCD2 (ID2) complex revealed that each paralog has an extensive α-solenoid fold contorted into a saxophone-like shape [18]. Interestingly, the ubiquitination target lysines are partially buried at the FANCI-FANCD2 interface, which extends throughout the N-terminal halves of the proteins. It has been suggested that the ubiquitin conjugated on FANCI interacts with FANCD2 [19]. The presence of DNA promotes ubiquitination of both isolated FANCI and ID2 complex *in vitro* [20,21], but it is currently unknown how this is achieved. While isolated FANCI and ID2 complex are well known to bind various DNA structures [18,20–23], isolated FANCD2 is less well established to bind DNA [18,22,24]. A FA patient mutation in FANCI, R1285Q, which reduces ubiquitination of the ID2 complex [20,21], has been suggested to reduce both FANCI and ID2 DNA binding, as well as FANCI interaction with FANCD2; however, the magnitude of reduction in DNA binding contrasts between the studies [20–22].

Although FANCD2 monoubiquitination has been documented for almost two decades [6] and FANCI monoubiquitination for over one decade [7], the molecular function of these modifications has been elusive. This has been largely due to the difficulty in isolating pure monoubiquitinated FANCI and FANCD2 proteins for *in vitro* studies. Recent advances in the understanding of the Ube2T allosteric

---

1 Institute of Molecular Cell and Systems Biology, College of Medical Veterinary and Life Sciences, University of Glasgow, Glasgow, UK
2 Scottish Centre for Macromolecular Imaging, University of Glasgow, Glasgow, UK
*Corresponding author. Tel: +44 141 3307212; E-mail: Helen.Walden@glasgow.ac.uk

activation by FANCL have allowed for the development of an engineered Ube2T which retains FANCI/FANCD2 lysine specificity but displays enhanced monoubiquitination activity [25]. This engineered Ube2T has facilitated preparation and isolation of highly purified ubiquitinated FANCI and FANCD2 without the need of DNA [26]. Here, we have used this approach to reconstitute the human ID2 complex in different states of ubiquitination and have characterized DNA binding for each state. We show that ubiquitination of FANCD2 significantly enhances binding of the ID2 complex to dsDNA, while ubiquitination of FANCI appears to be dispensable for this purpose. CryoEM maps of ubiquitinated FANCD2 in complex with either FANCI, or ubiquitinated FANCI and dsDNA, demonstrate a closure of the ID2 complex via formation of a new protein–protein interface at the C-termini. This interface is apparently disrupted in the FANCI R1285Q pathogenic mutant. We further demonstrate that ubiquitination of FANCI largely protects the ID2 complex from USP1-UAF1 deubiquitination, which likely contributes to the maintenance of ubiquitination-associated ID2-DNA binding enhancement in the cellular context. Therefore, it appears that ubiquitination of FANCI and FANCD2 have separate functions but converge to facilitate and maintain improved ID2-DNA binding.

## Results and Discussion

### Ubiquitination of FANCD2 enhances ID2-dsDNA binding

In order to explore whether FANCI and FANCD2 ubiquitination impacts ID2-DNA binding, we first ubiquitinated and purified human FANCI ($I_{Ub}$) and FANCD2 ($D2_{Ub}$) separately using our previously established protocol, which does not require the use of DNA [25,26]. We then reconstituted the non-ubiquitinated ID2 complex (I + D2), the $ID2_{Ub}$ complex, with ubiquitin only on FANCD2 (I + $D2_{Ub}$), and the $I_{Ub}D2_{Ub}$ complex, with ubiquitin on both FANCI and FANCD2 ($I_{Ub}$ + $D2_{Ub}$). We employed both solution-based protein-induced fluorescence enhancement (PIFE; 22°C) [27] and gel-based electro-mobility shift assays (EMSAs; 4°C) to assess dsDNA binding to the above complexes (32 base pair, IRDye700 labelled; Figs 1 and EV1). The two techniques revealed a striking enhancement of DNA binding when FANCD2 was ubiquitinated ($ID2_{Ub}$) compared to the unmodified complex (ID2). However, we observed that ID2 and $ID2_{Ub}$ DNA binding in PIFE experiments was highly sensitive to salt concentration (Fig EV1A), as expected for DNA–protein interactions. In contrast, corresponding dissociation constants determined by EMSA were only modestly affected due to salt changes and were significantly lower than PIFE (Fig EV1B). We reasoned that the above were due to protein–DNA samples entering a virtually salt-free gel environment (0.5× TBE) in the case of EMSA and hence this technique may not result in dissociation constants that reflect the salt environment where binding takes place. Thus, we determined apparent dissociation constants at physiological NaCl concentrations (150 mM) using PIFE (Fig 1A).

A ~ 10-fold enhancement of DNA binding affinity was observed for $ID2_{Ub}$ compared to ID2 (Fig 1A). DNA binding was not detectable under similar concentrations of isolated $D2_{Ub}$, suggesting that the ubiquitin on FANCD2 is not required for DNA binding *per se*, but enables a stronger ID2-DNA interaction. Interestingly, the di-monoubiquitinated complex ($I_{Ub}D2_{Ub}$) did not have a significantly different dsDNA binding affinity compared to $ID2_{Ub}$ (Fig 1A).

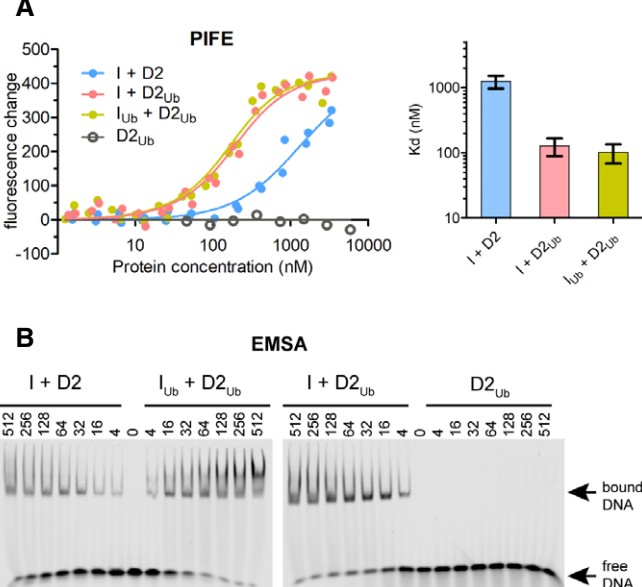

**Figure 1.  FANCD2 ubiquitination enhances ID2-dsDNA binding.**

IRDye700-labelled 32 base pair dsDNA was used to assess ID2-DNA binding, when neither protein is ubiquitinated (I + D2), when only FANCD2 is ubiquitinated (I + $D2_{Ub}$) and when both FANCD2 and FANCI are ubiquitinated ($I_{Ub}$ +$D2_{Ub}$).

A   *Left*: Fluorescence changes of IRDye700-labelled dsDNA (at 125 nM) when incubated at increasing ID2 (I + D2, $I_{Ub}$ + $D2_{Ub}$, I + $D2_{Ub}$) or ubiquitinated FANCD2 ($D2_{Ub}$) concentrations (ranging from 1.3 nM to 5.9 µM). Measurement of fluorescence enhancement for each ID2 complex was conducted for two separately prepared complexes (two technical repeats) and all data points for each protein combination were used in fitting of a one-site binding model. *Right*: Bar graph showing mean apparent $K_d$ values calculated from model fitting. Error bars: Asymmetric 95% confidence intervals from non-linear regression (23–24 data points each).

B   Assessment of protein–DNA interactions using electro-mobility shift assays (EMSAs). IRDye700-labelled dsDNA (at 2 nM) was incubated with indicated amounts of non/single/double-ubiquitinated ID2 ($His_6$-TEV-V5-FANCI and FLAG-FANCD2) protein complexes (I + D2, $I_{Ub}$ + $D2_{Ub}$, I + $D2_{Ub}$) or ubiquitinated FLAG-FANCD2 ($D2_{Ub}$). Mixes were run on non-denaturing gels, and the resolved free- and protein-bound DNA bands were visualized using an infrared scanner. EMSA gels of ID2 complexes are representative of 3 replicate experiments.

Source data are available online for this figure.

Similarly, EMSAs of reconstituted complexes also showed that $ID2_{Ub}$ and $I_{Ub}D2_{Ub}$ binding to dsDNA was comparable, and enhanced relative to ID2, whereas $D2_{Ub}$ showed no DNA binding (Fig 1B). These data suggest that FANCD2 ubiquitination serves to either promote or stabilize ID2-DNA binding and are consistent with associated increases in ID2-DNA binding upon FANCD2 ubiquitination reported in three recent EMSA-based studies [28–30].

### Ubiquitination of FANCD2 is associated with formation of a secondary ID2 interface

To examine the structural details of enhanced DNA binding affinity, we determined cryoEM maps of reconstituted human ID2 complexes with ubiquitinated FANCD2, at modest resolutions (Figs 2A and

EV2). Reference-free 2D class averages of $ID2_{Ub}$ and $I_{Ub}D2_{Ub}$-dsDNA exhibited similar overall shapes, but different to previous non-ubiquitinated ID2 class averages [12], hinting at a gross conformational change upon ubiquitination of FANCD2. Reconstructed maps of $ID2_{Ub}$ and $I_{Ub}D2_{Ub}$-dsDNA, at resolutions of 24 and 12 Å, respectively, both exhibited a closed, torus-like shape. Fitting of the mouse truncated ID2 crystal structure (3S4W) into the $I_{Ub}D2_{Ub}$-dsDNA map resulted in a poor fit (Fig 2B; *left panel*), but flexible fitting using iMODFIT [31] with secondary structure restraints improved the agreement (cross-correlation score from 0.56 to 0.85). The primary movement occurred for the FANCD2 C-terminal "arm" and resulted in formation of a new interface with the FANCI C-terminal "arm" that closes the ID2 complex (Fig 2B; *right panel*). We refer to this interface henceforth as the Arm ID2 interface. A difference map between the fitted model and the experimental map illustrates a tube-like volume, most likely representing the bound DNA. This volume is positioned just below the Arm ID2 interface and encompassed within the torus (Fig 2B; *right panel*), suggesting that formation of the Arm ID2 interface is necessary for this binding conformation. At these resolutions, we are not able to unambiguously place the conjugated ubiquitins. We propose that the closed conformation must be stabilized to tightly bind DNA and ubiquitination of FANCD2 acts for this purpose. The closed conformation is consistent with recent cryoEM structures of DNA-bound human $ID2_{Ub}$ and $I_{Ub}D2_{Ub}$ [28], and chicken $ID2_{Ub}$ [29].

### The Arm ID2 interface is required for enhanced ID2-dsDNA binding

Interestingly, the site of the pathogenic FANCI mutation R1285Q is in proximity to the Arm ID2 interface (Fig 3A) and in the recent atomic model of human $I_{Ub}D2_{Ub}$ forms a salt bridge with Q1365 on FANCD2 [28]. We hypothesized that this mutation may disturb ID2 ubiquitination by reducing formation of the closed ID2 state. We first examined whether this mutation brings any changes in FANCI's capacity to interact with DNA and FANCD2, as well as whether it affects FANCI's ability to get ubiquitinated. We found that both wild-type ($I^{WT}$) and mutant ($I^{R1285Q}$) proteins could be ubiquitinated *in vitro* to the same extent, and addition of DNA resulted in comparable enhancement of ubiquitination between the two proteins (Fig 3B). Furthermore, by measuring the binding affinities of RED-tris-NTA (NanoTemper) labelled His-tagged $I^{WT}$ and $I^{R1285Q}$ for FANCD2 (using PIFE), we found that the affinities were similar and both in the low nanomolar range (Fig 3C). Nevertheless, the FANCI mutation resulted in an apparent reduction in FANCD2 ubiquitination in the ID2 complex (Fig 3D), consistent with previous results [20,21,28]. Under our assay conditions, we did not detect a significant change in FANCI ubiquitination in the ID2 complex due to the mutation. Nevertheless, the FANCI R1285Q mutation was recently shown to result, not only in a reduction of FA core catalysed FANCI and FANCD2 ubiquitination within an ID2 complex [preprint: 14], but also in faster deubiquitination of the ubiquitinated complex [28]. This slower ubiquitination and faster deubiquitination may explain the nearly complete absence of ubiquitinated FANCD2/FANCI seen in cells having the FANCI R1285Q mutation [7].

Interestingly, the reconstituted mutant $I^{R1285Q}D2_{Ub}$ complex behaved differently in terms of dsDNA binding enhancement, compared to the wild-type $ID2_{Ub}$ complex. PIFE revealed only a minor, insignificant reduction of ID2-DNA affinity when FANCI was mutated to $FANCI^{R1285Q}$, and when FANCD2 was ubiquitinated the ID2-DNA affinity was not substantially enhanced, unlike that seen for wild-type complex (Fig 3E). EMSAs similarly showed a small increase in $I^{R1285Q}D2$-DNA binding when FANCD2 was ubiquitinated (Fig 3F), unlike the levels observed with $I^{WT}D2$ versus $I^{WT}D2_{Ub}$. Taken together, these results suggest that the $FANCI^{R1285Q}$ patient mutation does not directly alter ID2-dsDNA binding, but instead restricts FANCD2 ubiquitination and the associated DNA binding enhancement. This is likely achieved via disruption of the Arm ID2 interface, seen in the closed ID2 state. Hence, the loss of FANCD2 ubiquitination in $I^{R1285Q}D2$ complex can be rationalized if the closed state is also important for its ubiquitination.

### FANCI ubiquitination protects $ID2_{Ub}$ complex against USP1-UAF1 deubiquitination

The USP1-UAF1 complex specifically targets ubiquitinated FANCD2 for deubiquitination utilizing an N-terminal module of USP1 [32]. Although this can occur when $D2_{Ub}$ is in isolation or in complex with FANCI, di-monoubiquitinated ID2 complexes ($I_{Ub}D2_{Ub}$) bound to DNA remain largely resistant to USP1-UAF1 deubiquitination [32,33]. We hypothesized that since FANCI ubiquitination does not further enhance ID2-DNA binding, its primary role may be in protecting FANCD2's ubiquitin from USP1-UAF1 activity. To examine to what extent the presence of I or $I_{Ub}$ influences $D2_{Ub}$ deubiquitination, we compared the progress of $D2_{Ub}$, $ID2_{Ub}$ and $I_{Ub}D2_{Ub}$ deubiquitination by USP1-UAF1 (in the presence of dsDNA) in a time course (Fig 4A). USP1-UAF1, at low (25 nM) concentrations, deubiquitinated both $D2_{Ub}$ and $ID2_{Ub}$ at similar rates. However, the $I_{Ub}D2_{Ub}$ substrate remained almost completely resistant to USP1-UAF1 activity (Fig 4A; *Left*). At higher concentrations of USP1-UAF1 (100 nM), we found that FANCD2 can be deubiquitinated, albeit at slower rate than $D2_{Ub}$ and $ID2_{Ub}$, which are rapidly deubiquitinated in under 10 min (Fig 4A; *Right*). These data suggest that access to the ubiquitin on FANCD2 for USP1-UAF1 is reduced when in complex with ubiquitinated FANCI and DNA. *In vitro* [25,33] and cell-based [7] assays have shown that the blockage of FANCI ubiquitination also results in reduced FANCD2 ubiquitination. According to our data, this may occur because FANCD2's ubiquitin is no longer sufficiently protected against USP1-UAF1 deubiquitination.

Since USP1-UAF1 is a specific deubiquitinase (DUB) for FANCD2 [15,32], we wanted to assess whether alternative DUBs, that lack FANCD2 specificity (i.e. target structurally diverse substrates), are also able to access the ubiquitin on FANCD2 when this is in complex with I or $I_{Ub}$. We therefore assayed deubiquitination of $D2_{Ub}$, $ID2_{Ub}$ and $I_{Ub}D2_{Ub}$ by USP7 and USP2 (in the presence of dsDNA) in a time course (Fig 4B). While USP7 and USP2 could deubiquitinate $D2_{Ub}$ in isolation, the presence of FANCI or ubiquitinated FANCI reduced $D2_{Ub}$ deubiquitination. This result suggests that, in the $ID2_{Ub}$-DNA and $I_{Ub}D2_{Ub}$-DNA complexes, the ubiquitin on FANCD2 is protected against generic DUB activity. A potential explanation for the protection is that FANCI may block DUB access by interacting with FANCD2's ubiquitin. Indeed, recently reported $ID2_{Ub}$-DNA structures show that FANCD2's ubiquitin directly interacts with FANCI [28,29]. However, USP1-UAF1 is apparently able to circumvent any protection formed from a locked $ID2_{Ub}$-DNA

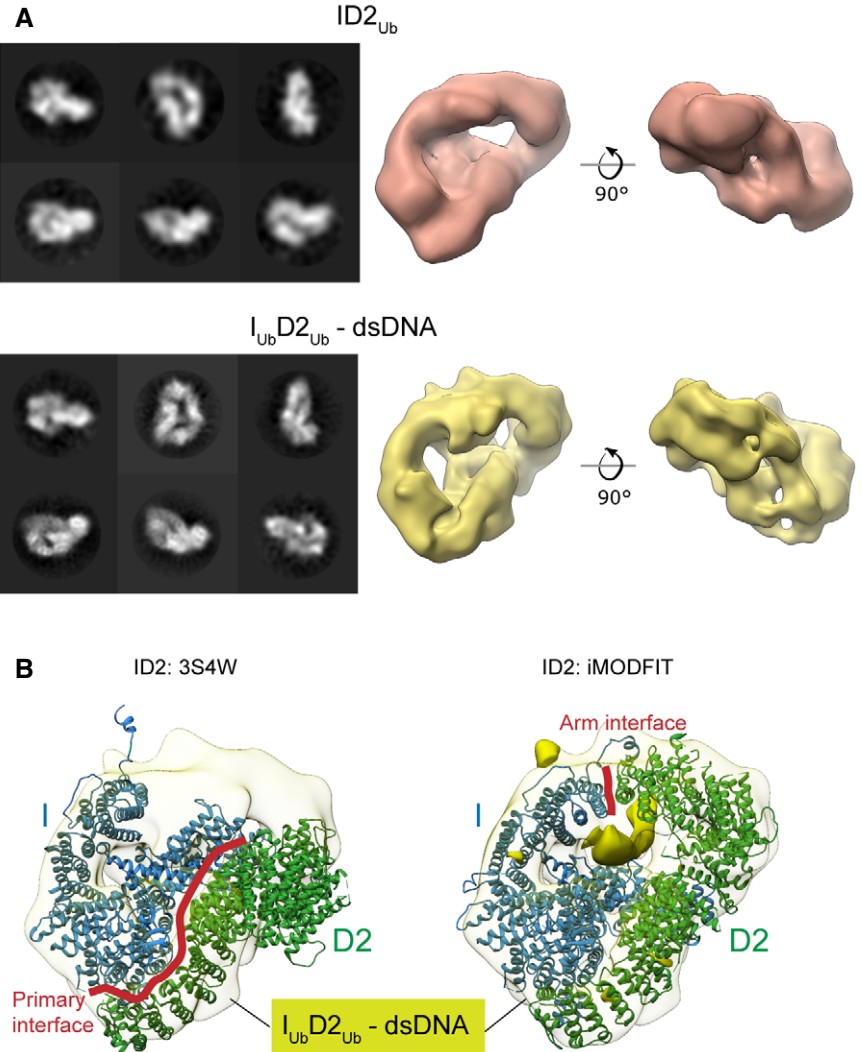

**Figure 2. FANCD2 ubiquitination creates a secondary ID2 "Arm" interface, resulting in a closed ID2 conformation.**

A  *Left*: CryoEM 2D classes of $ID2_{Ub}$ and $I_{Ub}D2_{Ub}$-dsDNA. *Right*: 3D reconstructions of $ID2_{Ub}$ (EMD-10843) and $I_{Ub}D2_{Ub}$-DNA (EMD-10844). Both structures exhibit a torus-like shape.

B  *Left*: Rigid body fit of the mouse ID2 atomic structure (3S4W) into the human $I_{Ub}D2_{Ub}$-dsDNA cryoEM map. *Right*: Flexible fit of the mouse ID2 atomic model into the human $I_{Ub}D2_{Ub}$-dsDNA cryoEM map (iMODFIT). The C-termini "arms" of FANCD2 and FANCI close in the flexible fitting. A tube-like volume in the $I_{Ub}D2_{Ub}$-DNA cryoEM map, which is not occupied by the ID2 iMODFIT structure, is present just beneath the Arm ID2 interface and likely represents bound DNA (solid density).

complex. These data suggest that USP1-UAF1 may be capable of modulating the $ID2_{Ub}$-DNA complex in order to gain access to FANCD2's conjugated ubiquitin. Although the exact details by which this is achieved are not known, our recent work has shown that a FANCD2-binding sequence, located at USP1's N-terminus, is required for efficient $ID2_{Ub}$-DNA deubiquitination [32]. Nevertheless, our work here shows that this ability of USP1-UAF1 is compromised when FANCI is also ubiquitinated. The effect of FANCI's ubiquitination may be on rendering the doubly ubiquitinated ID2 complex unamenable to USP1-UAF1 modulation, for example by stabilizing an ID2 conformational change induced by FANCD2 ubiquitination or inducing further minor conformational changes that stabilize the complex. Alternatively, FANCI ubiquitination may directly disrupt USP1-UAF1 binding to FANCD2.

One explanation for FANCI's conjugated ubiquitin facilitating a USP1-UAF1 resistant complex is that this ubiquitin interacts with FANCD2. Ubiquitin typically interacts with other proteins via any of its three (F4, I36 and I44) hydrophobic patches [34]. Thus, we mutated four key hydrophobic residues of ubiquitin located within these patches, which are at distinct regions of the ubiquitin structure (Fig 4C). Subsequently, we ubiquitinated FANCI (in the presence dsDNA) using either wild-type or one of these four ubiquitin mutants (F4A, I36A, L73A or I44A). Nearly complete FANCI ubiquitination was achieved for each mutant and wild-type ubiquitin (Fig 4D). We tested each $I_{Ub}$ in DUB assays and found that only $I_{Ub-I44A}$ was also deubiquitinated to the same extent as $I_{Ub-WT}$ by USP1-UAF1. The I36A mutation resulted in partial loss, whereas the F4A and L73A mutations on ubiquitin resulted in complete loss of USP1-

UAF1 activity (Fig 4D). To assemble $I_{Ub}D2_{Ub}$-DNA complex with ubiquitin mutants on FANCI, we added $D2_{Ub}$ to each ubiquitin-mutated $I_{Ub}$ and subjected the resulting $I_{Ub}D2_{Ub}$-DNA complex to deubiquitination treatment with USP1-UAF1 (Fig 4E). Both F4A and I44A ubiquitin mutations, despite having contrasting effects on deubiquitination of FANCI alone, resulted in an increased suscepti-bility to FANCD2 deubiquitination. The I36A and L73A mutations

had negligible effects on FANCD2 deubiquitination (Fig 4E). The same effect of I44A and F4A on FANCD2 deubiquitination was observed when using four times lower concentrations of USP1-UAF1 in single time-point DUB assays (Fig EV3A). We reasoned that ID2-DNA binding or complex formation was unaffected by these mutations, since WT, F4A and I44A $I_{Ub}D2_{Ub}$ complexes were still able to bind DNA efficiently, unlike $I_{Ub}$ alone (Fig EV3B) or $D2_{Ub}$

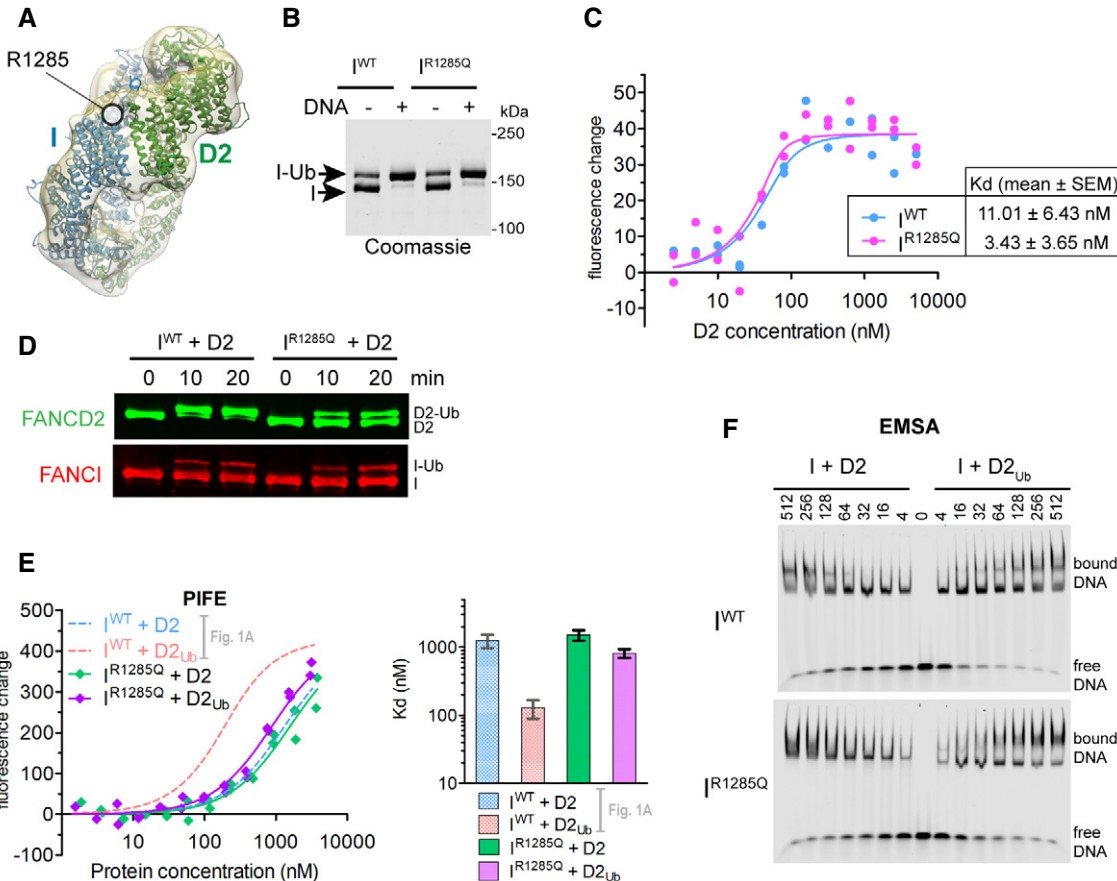

**Figure 3. The pathological FANCI R1285Q mutation disrupts ID2 ubiquitination, likely via disturbing the closed ID2 conformation.**

A Arginine 1285 of FANCI, which is mutated in some FA patients, is located in the Arm ID2 interface in our cryoEM $I_{Ub}D2_{Ub}$-dsDNA structure.

B Both wild-type ($I^{WT}$) and R1285Q mutant ($I^{R1285Q}$) FANCI can be efficiently ubiquitinated in the presence of DNA. Reactions were performed at 2 μM FANCI substrate for 60 min in the absence or presence of 4 μM ssDNA.

C Both wild-type ($I^{WT}$) and R1285Q mutant ($I^{R1285Q}$) FANCI efficiently associate with FANCD2. Fluorescence changes occurring when RED-tris-NTA-labelled FANCI ($I^{WT}$ or $I^{R1285Q}$; both at 60 nM) is incubated at increasing concentrations of FANCD2 (ranging from 2.48 nM to 5.08 μM). Titrations were conducted twice (two technical replicates), and all data points for each protein combination were used in fitting of a one-site binding model. Apparent $K_d$ values (mean ± SEM) derived from fitting of a one-site binding model to the 24 data points of the two technical replicates are shown.

D FANCD2 within an $I^{R1285Q}$D2 complex is resistant to ubiquitination. Reactions were performed at 4 μM ID2 substrate and 16 μM dsDNA. Progress of FANCD2 and FANCI ubiquitination was monitored by Western blotting following SDS–PAGE.

E FANCD2 ubiquitination cannot robustly enhance ID2-DNA binding when complexed with $I^{R1285Q}$, as determined by PIFE. *Left*: Fluorescence changes when IRDye700-labelled dsDNA (at 125 nM) is incubated at increasing concentrations of $I^{R1285Q}$ + D2 or $I^{R1285Q}$ + $D2_{Ub}$ (complex concentrations ranging from 1.54 nM to 3.8 μM). Measurement of fluorescence enhancement for each protein combination was conducted for two separately prepared complexes (two technical repeats), and all data points for each protein combination were used in fitting of a one-site binding model. *Right*: Bar graph showing mean apparent $K_d$ values calculated from the one-site binding model. Error bars: Asymmetric 95% confidence intervals from non-linear regression (22 data points each). $I^{WT}$ + D2 and $I^{WT}$ + $D2_{Ub}$ previously calculated curves and corresponding $K_d$ values (from data points shown in Fig 1A) are also shown for comparison.

F FANCD2 ubiquitination cannot robustly enhance ID2-DNA binding when complexed with $I^{R1285Q}$, as determined by EMSAs. IRDye700-labelled dsDNA (at 2 nM) was incubated with indicated amounts of ID2 (His$_6$-FANCI or His$_6$-FANCI$^{R1285Q}$) mixed with non-ubiquitinated or ubiquitinated FLAG-FANCD2. Mixes were run on non-denaturing gels, and the resolved free- and protein-bound DNA bands were visualized using an infrared scanner. Gels shown are representative of two replicate experiments.

Source data are available online for this figure.

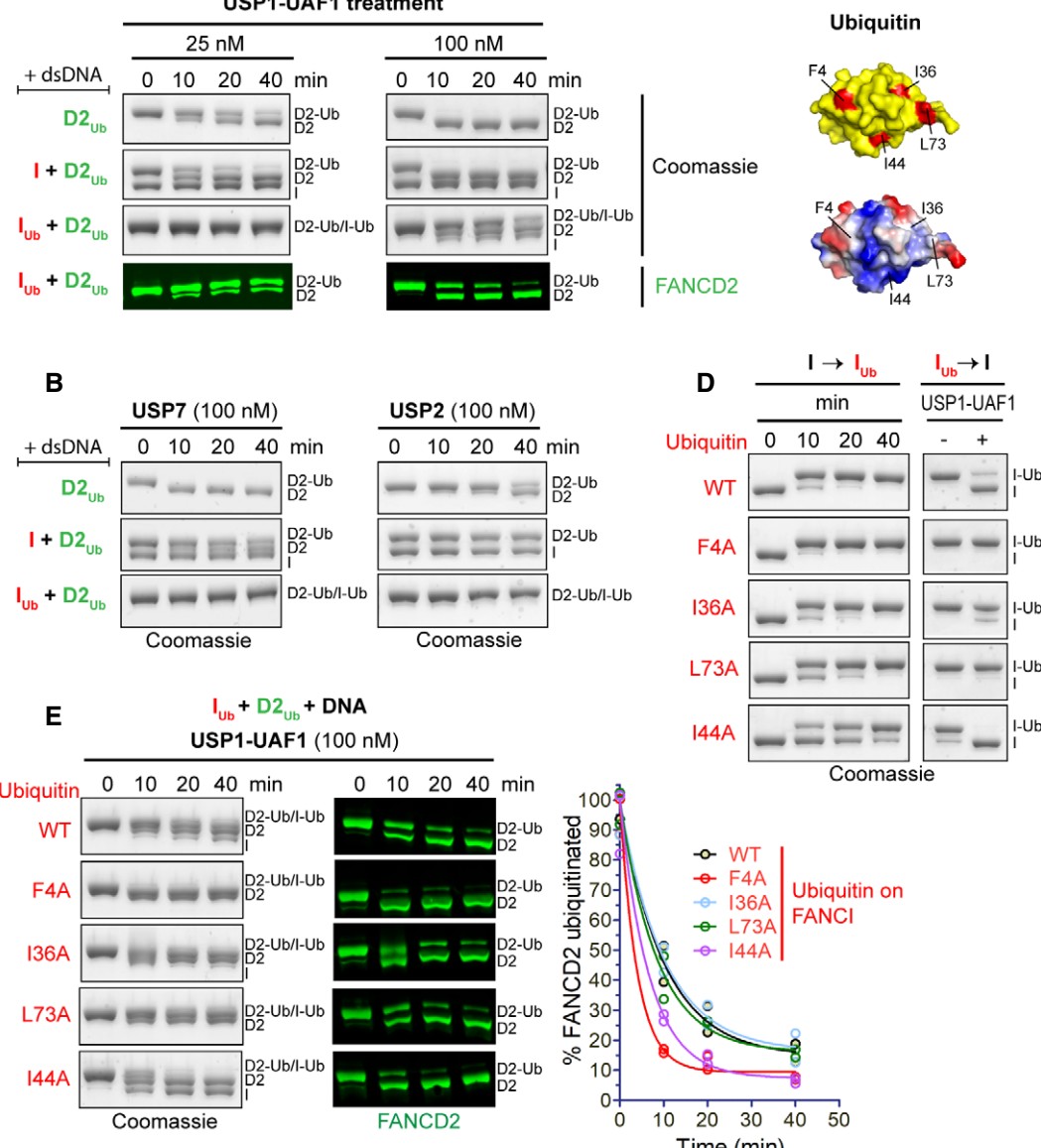

**Figure 4. FANCI ubiquitination of ID2$_{Ub}$ protects D2$_{Ub}$ from USP1-UAF1-mediated deubiquitination.**

A   USP1-UAF1 can efficiently deubiquitinate D2$_{Ub}$ in the presence or absence of FANCI (I), but not in the presence of ubiquitinated FANCI (I$_{Ub}$). Ubiquitinated FANCD2 (D2$_{Ub}$) was mixed with a 50 base pair dsDNA and either His$_6$-TEV-V5-FANCI (I), ubiquitinated His$_6$-TEV-V5-FANCI (I$_{Ub}$) or no protein; protein–DNA mixes were subsequently incubated with USP1-UAF1 (at 25 nM or 100 nM) for indicated time periods. Deubiquitination of D2$_{Ub}$ and I$_{Ub}$ was assessed, following SDS–PAGE, by Coomassie staining of the gels, as well as by Western blotting of transferred blots with a specific FANCD2 antibody.

B   USP7 and USP2 can deubiquitinate D2$_{Ub}$, but their activity towards D2$_{Ub}$ is greatly reduced in the presence of FANCI (I) or ubiquitinated FANCI (I$_{Ub}$). Reactions were set up as in (A), but with 100 nM USP7 or USP2. Deubiquitination was assessed, following SDS–PAGE, by Coomassie staining.

C   Location of the four hydrophobic residues mutated to alanine in the ubiquitin structure (PDB: 1ubq). *Top*: location of F4, I36, I44 and L73 (shown in red) in ubiquitin's surface (shown in yellow). *Bottom*: same as above but with ubiquitin's surface coloured according to charge (red: negative, blue: positive, white: no charge).

D   Time course of FANCI ubiquitination with various ubiquitin mutants and corresponding sensitivity/resistance to USP1-UAF1-mediated deubiquitination of resulting products. Deubiquitination reactions in the presence or absence of USP1-UAF1 (100 nM) incubated for 30 min.

E   Time course of USP1-UAF1-mediated deubiquitination of I$_{Ub}$D2$_{Ub}$-DNA complexes consisting of D2$_{Ub}$ a 50 base pair dsDNA and I$_{Ub}$ produced with various ubiquitin mutants or wild-type (WT) ubiquitin. Progress of deubiquitination reaction was assessed following SDS–PAGE, by both Coomassie staining of the gels and Western blotting of transferred blots with a specific FANCD2 antibody. FANCI ubiquitination with indicated ubiquitin mutants (or WT) and subsequent deubiquitination of resulting I$_{Ub}$D2$_{Ub}$-DNA complexes were conducted twice; the residual FANCD2 ubiquitination, calculated from the FANCD2 blots for each time-point, was plotted for each type of ubiquitin in the protein complex, and corresponding deubiquitination progress curves were fitted using a one-phase decay model.

alone (Fig 1). Taken together, these data suggest the F4 and I44 hydrophobic surfaces on ubiquitin are important for $I_{Ub}$-mediated protection of $D2_{Ub}$, potentially via interaction with FANCD2. Indeed, in the recently reported $I_{Ub}D2_{Ub}$-DNA structure [28], I44 of FANCI's conjugated ubiquitin is located within the FANCD2–ubiquitin interface. In contrast, F4 does not appear to directly contact FANCD2 in this structure. Hence, the F4A mutation may instead favour FANCD2 deubiquitination either by disturbing FANCI deubiquitination (Fig 4D), or via changing ubiquitin dynamics [35], thus altering the ubiquitin–FANCD2 interface.

Interestingly, the ubiquitin on FANCI was also protected from USP2, USP7 and USP1-UAF1 deubiquitination in the $I_{Ub}D2_{Ub}$-DNA complex. In fact, in the context of the $I_{Ub}D2_{Ub}$-DNA complex, FANCI deubiquitination was far from complete at 40 min, even at the highest USP1-UAF1 concentration used (Fig EV4). However, isolated $I_{Ub}$ was completely deubiquitinated within 30 min by USP1-UAF1 (Fig 4D). Moreover, we observed that FANCD2 was deubiquitinated by USP1-UAF1 faster than FANCI in the $I_{Ub}D2_{Ub}$ complex (Fig EV4). This observation implies that an ordered deubiquitination mechanism may be in place, resulting in an $I_{Ub}D2$ complex intermediate. Our reconstitution approach allowed us to test the DNA binding affinity of this complex, which was found to be intermediate, i.e. tighter than that of ID2 but weaker than that of $I_{Ub}D2_{Ub}$ (Fig EV5). Hence, if such a complex results from USP1-UAF1 deubiquitination, it is likely to retain a certain extent of the strong association with DNA observed with $I_{Ub}D2_{Ub}$, potentially via partially stabilizing the closed state.

Taken together, our results suggest that the sequential action of FANCD2 and FANCI ubiquitination results in a stable $I_{Ub}D2_{Ub}$-DNA complex, resistant to deubiquitination. The deubiquitination resistance of this complex likely serves to increase the *in vivo* half-life of the closed ID2 complex. In that way, a threshold event is created whereby once two ubiquitins are installed on the ID2 complex the latter is committed for its function. In contrast, the intermediate $ID2_{Ub}$ complex is transient and rapidly deubiquitinated.

Previous work has shown that when DNA is removed from the reaction, di-monoubiquitinated ID2 is no longer resistant to deubiquitination by USP1-UAF1 [32,33], but it is currently not clear how DNA may protect $I_{Ub}D2_{Ub}$ from USP1-UAF1 deubiquitination. One possibility is that in the absence of DNA ubiquitinated FANCD2 dissociates from ubiquitinated FANCI, and in isolation both proteins are known to be susceptible to USP1-UAF1 activity [32]. However, the effect of DNA binding on the strength of interaction between FANCI and FANCD2 in different ubiquitination states is unknown and will require description beyond 1:1 binding models. Nevertheless, removing DNA might be an effective way of achieving rapid ID2 deubiquitination and switching off the pathway. This could be achieved in a cellular context by extraction of ubiquitinated ID2 complexes by the DVC1-p97 segregase [36] and subsequent deubiquitination by USP1-UAF1. Future work will be required to understand the sequence of events, DNA dependence and when other factors are involved in $I_{Ub}D2_{Ub}$ deubiquitination.

### The combined action of FANCD2 and FANCI ubiquitination stabilizes ID2 on DNA

Our above results demonstrate separate functions for the two monoubiquitination events occurring on the ID2 complex. FANCD2

ubiquitination was found to modulate ID2-dsDNA binding affinity, whereas FANCI ubiquitination was found to largely protect the ubiquitinated ID2 complex from deubiquitination. A simple explanation for our data is that the ID2 complex can explore two different conformational extremes: an open state, as in the previously reported non-ubiquitinated ID2 crystal structure and EM maps [12,18,37], and a closed state, as observed here (Fig 5). We propose that DNA binding allows a population of ID2 to reach the closed ID2 conformation, in which the Arm ID2 interface forms. This conformational change likely exposes FANCD2's target lysine, as well as its adjacent acidic patch [25] to allow docking of Ube2T and subsequent FANCD2 ubiquitination, which stabilizes the closed state. However, the action of USP1-UAF1 within the cell will allow a population of $ID2_{Ub}$ to revert to an open state via the unstable closed state of deubiquitinated ID2. The sequential ubiquitination of FANCI may ensure that the majority of the ID2 population exists in the closed state, since $I_{Ub}D2_{Ub}$ is largely protected from USP1-UAF1 activity. Hence, FANCD2 ubiquitination appears to enhance ID2-DNA binding by reaching the closed state, while FANCI ubiquitination acts to maintain this closed state (and thus the higher DNA affinity), by impairing deubiquitination. As a result, the sequential action of FANCD2 and FANCI ubiquitination is expected to lock the ID2 complex on DNA. Since FANCI's Arginine 1285 is located near the observed Arm ID2 interface in our cryoEM structures, the R1285Q mutation on FANCI is expected to disturb this interface by inhibiting the closed ID2 state and subsequent ubiquitination events that depend on this (Fig 5). Indeed, R1285 of FANCI, although unstructured in non-ubiquitinated ID2, was recently shown to form a salt bridge with E1365 of FANCD2 in an $I_{Ub}D2_{Ub}$-DNA structure [28].

Our model suggests a sequence of events, where a FANCD2 ubiquitination is first required to achieve a shift towards an ID2 state with higher DNA affinity and then FANCI ubiquitination follows to maintain this state. This is consistent with the observations that FANCI ubiquitination lags behind FANCD2 ubiquitination *in vitro* [25,33]. Although we cannot exclude the possibility of an $I_{Ub}D2$ complex population existing *in vivo*, such a population is more likely to arise from $I_{Ub}D2_{Ub}$ deubiquitination (Fig EV4). The $I_{Ub}D2$ complex apparently has partially enhanced ID2-DNA affinity, and in light of the recent ID2 and $I_{Ub}D2_{Ub}$ structures where the ubiquitin on FANCI clashes with FANCD2 in the unmodified state [28], there are likely to be some structural changes associated with $I_{Ub}D2$ complex compared to ID2.

It is possible that ubiquitination-dependent locking of ID2 on DNA ensures that the complex is properly recruited to sites of replication arrest where it may be needed. As such, our data are consistent with the observed enrichment of ubiquitinated forms of FAND2/FANCI in chromatin [7,8]. DNA-bound $ID2_{Ub}$ or $I_{Ub}D2_{Ub}$ complexes may be able to slide, recognize a specific DNA structure or execute another unknown function, which will result in the concentrated ID2 foci frequently observed in nuclei of DNA-damaged cells [7]. Recent reports indicate that $I_{Ub}D2_{Ub}$ is able to slide on circular DNA, since it dissociates from this much slower than linear DNA [28], and its association with dsDNA may result in filament-like structures [30].

Future work should seek to further validate the conformational changes driven by FANCI/FANCD2 ubiquitination in solution and the role of DNA in these changes. For example, SAXS measurements

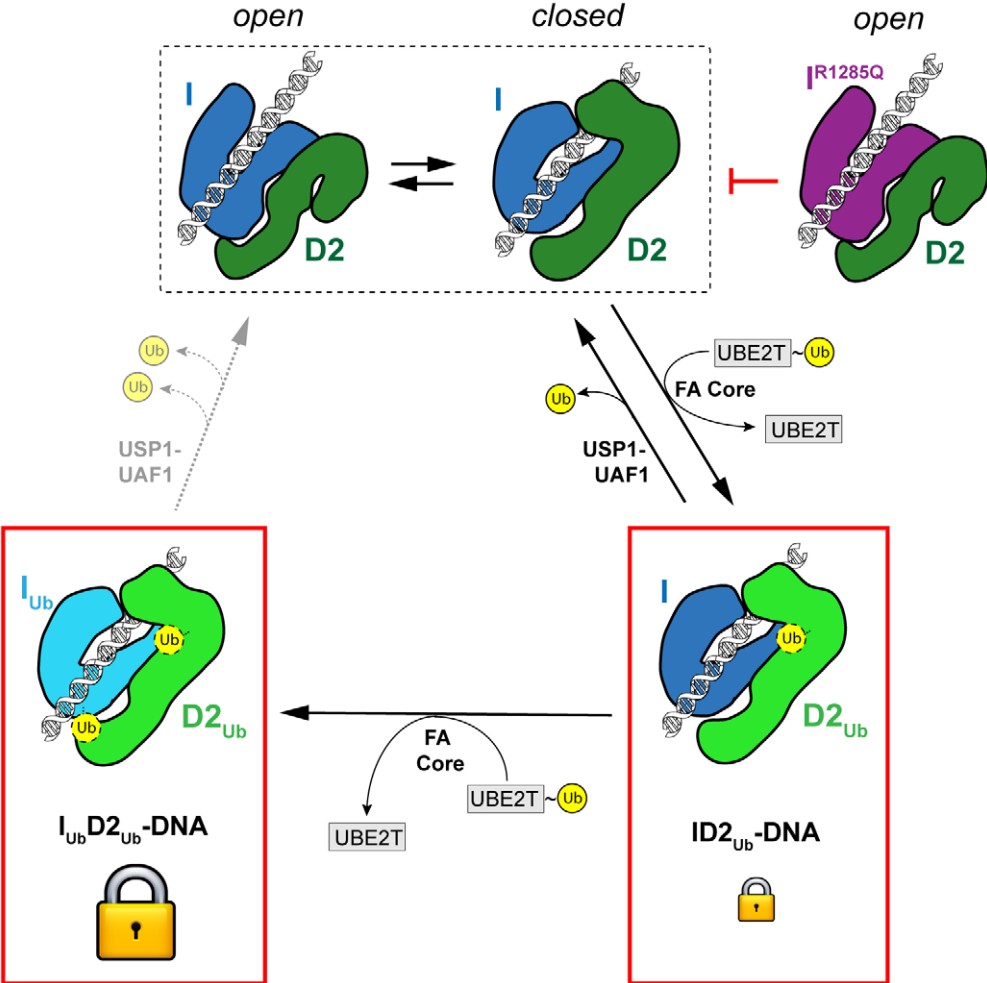

**Figure 5. Model of how di-monoubiquitination may shift the ID2 conformation to an enhanced DNA-binding state.**

ID2 interaction with DNA is proposed to promote a dynamic equilibrium where ID2 can exist in both an open and a closed conformation. FANCD2 ubiquitination by Ube2T and the FA core complex, within a closed ID2 conformation status, can shift this equilibrium in favour of the closed conformation, with the action of USP1-UAF1 counteracting such shift. When FANCI is also ubiquitinated by Ube2T and the FA core complex, both ubiquitins are largely resistant to USP1-UAF1 deubiquitination, and thus, the doubly ubiquitinated ID2 can remain in the closed conformation. The latter conformation has a tighter DNA affinity and hence locks ID2 onto DNA. The R1285Q mutation on FANCI likely restricts formation of a closed ID2 conformation, which in turn negatively impacts on ID2 ubiquitination and locking onto DNA.

of the different ubiquitination states could be compared. FRET measurements with FANCI and FANCD2, labelled near the C-termini, may be able to reveal the population of open and closed conformation for each ubiquitination state and how these populations are affected by DNA binding. In addition, cell-based experiments may be used to test the individual function of FANCI and FANCD2 ubiquitination *in vivo*. For example, regarding FANCD2 ubiquitination, the extent of ID2 chromatin association or foci formation could be assessed in cells where FANCI cannot be ubiquitinated and USP1-UAF1 is additionally defective.

Here, we have uncovered distinct functions for two specific monoubiquitination events on FANCI and FANCD2 within the ID2 complex. We find that FANCD2 ubiquitination enhances binding of the ID2 complex to dsDNA, while FANCI ubiquitination protects the complex from deubiquitination by USP1-UAF1. Combined, both events lead to a stable ID2 complex on dsDNA.

# Materials and Methods

**Cloning and mutagenesis of expression constructs**

Constructs encoding for human FANCD2 having an N-terminal 3C-cleavable His$_6$-tag (His$_6$-3C-FANCD2), and human FANCI having N-terminal His$_6$ and V5 tags separated by a TEV-cleavable site (His$_6$-TEV-V5-FANCI), were described previously [26]. Related *FANCD2* and *FANCI* constructs encoding for a His$_6$-3C-FLAG-tagged FANCD2 or His$_6$-FANCI proteins were produced by site-directed mutagenesis. The R1285Q mutation was introduced into His$_6$-FANCI by site-directed mutagenesis in the respective *FANCI* construct. Human *USP7* was ligated into a pFBDM vector by restriction cloning to encode for His$_6$-3C-USP7. N-terminally His$_6$-TEV-tagged USP1 (with G670A/G671A mutated auto-cleavage site [38]) and UAF1 were produced via sequential insertion of human *USP1* and *UAF1* cDNAs

into an appropriate pFBDM vector by restriction cloning. All other constructs have been reported previously [25,26,32]. The coding regions of all constructs were verified by DNA sequencing.

**Protein expression and purification**

All FANCI and FANCD2 constructs were expressed in *Sf*21 insect cells, lysed by sonication in the presence of benzonase and purified by Ni-NTA affinity chromatography, anion exchange and gel filtration as previously described [26]. In some cases, the His$_6$-tag was removed from His$_6$-3C-FANCD2 or His$_6$-3C-FLAG-FANCD2 by 3C protease cleavage, prior to the gel filtration step. Ubiquitinated FANCI and FANCD2 were produced and purified following *in vitro* reactions with Spy-3C-tagged ubiquitin, covalent linkage of ubiquitinated proteins with GST-tagged SpyCatcher, capture of resulting products on glutathione beads and subsequent cleavage of ubiquitinated proteins by 3C-protease treatment, as described previously [26]. For ubiquitinated His$_6$-TEV-V5-FANCI and FLAG-FANCD2 used in EMSAs, the steps including the reaction with Spy-3C-tagged ubiquitin and subsequent covalent linkage of ubiquitinated proteins with GST-SpyCatcher were instead replaced by reactions with GST-3C-tagged ubiquitin. The 3C-protease treatment of ubiquitinated FANCD2 also resulted in removal of the six-histidine-tag from ubiquitinated FANCD2 constructs. After the final gel filtration step, purified proteins in GF buffer (20 mM Tris pH 8.0, 400 mM NaCl, 5% glycerol, 5 mM DTT or 0.5 mM TCEP) were cryo-cooled in liquid nitrogen in single use aliquots. The His$_6$-FANCI R1285Q mutant was expressed and purified in the same way as the wild-type one.

Preparation of bacmids and protein expression in *Sf*21 insect cells for USP1-UAF1 and USP7 was performed as previously described for FANCI and FANCD2 [26]. All steps following cell harvesting and prior to protein storage were performed at 4°C. Cells were harvested ~ 72 h following baculovirus infection. Cells were pelleted by centrifugation, and cell pellets were resuspended in fresh ice-cold lysis buffer (50 mM Tris pH 8.0, 150 mM NaCl, 5% glycerol, 5 mM β-mercaptoethanol, 10 mM imidazole, EDTA-free protease inhibitor cocktail (Pierce), 2 mM MgCl$_2$ and benzonase). Cells were lysed by sonication and clarified (32,000 × g for 45 min) before proteins were bound to Ni-NTA resin. Ni-NTA-bound His$_6$-TEV-tagged USP1-UAF1 complex and His$_6$-3C-tagged USP7 were further washed with 50 mM Tris pH 8.0, 150 mM NaCl, 5% glycerol, 1 mM TCEP and 10 mM imidazole and then eluted into low salt buffer (50 mM Tris pH 8.0, 100 mM NaCl, 5% glycerol, 1 mM TCEP) containing 250 mM imidazole. Anion exchange was performed for both USP1-UAF1 and USP7 by binding proteins to a ResourceQ (1 ml) column and eluting over a linear gradient (20 column volumes) of NaCl (100–1,000 mM) in 20 mM Tris pH 8.0, 5% glycerol and 1 mM TCEP. Fractions containing USP1-UAF1 were subject to His-TEV protease overnight (ratio 1:10 protease to tagged protein). His$_6$-TEV-tagged USP1 and His$_6$-TEV protease were bound to Ni-NTA resin, and cleaved USP1-UAF1 complex was collected in the flow-through. In order to remove excess USP1, cleaved USP1-UAF1 was then concentrated and further purified using GL 10/300 Superdex 200 Increase column in 20 mM Tris pH 8.0, 150 mM NaCl, 5% glycerol and 5 mM DTT. Fractions from the peak containing the protein of interest were concentrated to ~ 5 mg/ml and cryo-cooled in liquid nitrogen as single use aliquots. For His$_6$-3C-tagged USP7, protein was concentrated and subject to two rounds of gel filtration using a

GL 10/300 Superdex 200 Increase column in 20 mM Tris pH 8.0, 150 mM NaCl, 5% glycerol and 5 mM DTT. The centre of the asymmetric gel filtration peak for USP7 was collected from the first run and purified again on gel filtration, and the final peak was symmetric and concentrated to ~ 5 mg/ml for flash freezing as single use aliquots in liquid nitrogen.

GST-USP2 was purified as previously described, and the GST tag was retained [32]. Protein ubiquitination reagents (Uba1, Ube2T/Ube2Tv4, FANCL$^{109-375}$) were prepared as described previously [26]. Protein concentrations were determined using absorbance at 280 nm and predicted extinction coefficients based on the protein sequences [39]. The 260/280 ratio of purified proteins measured to be < 0.8 for all samples, typically 0.6–0.65 suggesting minimal DNA content.

**DNA substrates**

IRDye700-labelled dsDNA (ds32$^F$) was obtained from annealing of two complementary 32 base pair HPLC-purified 5′-end IRDye700-labelled DNA ssDNA molecules. Non-labelled dsDNA (ds32 & ds50) was obtained from annealing of two PAGE-purified complementary (32-nucleotide-long or 50-nucleotide-long) ssDNA molecules. The 64-nucleotide-long ssDNA (ss64) was similarly PAGE-purified. The above dsDNAs were purchased from Integrated DNA Technologies in their final purified and/or annealed form, were subsequently resuspended at appropriate stock concentrations in distilled water and stored at −20°C until use. ssDNA was purchased from Sigma Aldrich, resuspended at appropriate stock concentrations in 10 mM Tris pH 8.0, 25 mM NaCl and 0.5 mM EDTA, and stored at −20°C until use. Sequence details of all oligonucleotides used are provided in Appendix Table S1.

**Assessment of FANCI/FANCD2 ubiquitination or subsequent deubiquitination by SDS–PAGE, and Coomassie staining or Western blotting**

Non-ubiquitinated/deubiquitinated FANCI or FANCD2 proteins were distinguished from respective ubiquitinated products/substrates following SDS–PAGE on Novex 4–12% Tris-glycine gels (Thermo Fisher) and subsequent staining of the gels with Instant-Blue Coomassie stain (Expedeon). All samples loaded for SDS–PAGE (~ 300 ng of FANCD2 for Coomassie staining and ~ 100 ng for Western blotting) were first diluted with reducing LDS buffer [consisting of NuPAGE 4× LDS buffer (Thermo Fisher) and appropriate concentration of beta-mercaptoethanol] to 1× LDS and 100 mM beta-mercaptoethanol or 100 mM DTT final, and then heated for 2 min at 100°C. For assessing the progress of deubiquitination, ubiquitinated/deubiquitinated FANCI and FANCD2 were additionally visualized by Western blotting. SDS–PAGE-separated proteins were transferred onto nitrocellulose membranes using an iBlot gel transfer device (Invitrogen) set at P0 (20 V 1 min, 23 V 4 min, 25 V 2 min) and blocked with 5% milk PBS-T (0.05% Tween 20) before incubation with 1:1,000 rabbit anti-FANCD2 (sc-28194; Santa Cruz Biotechnology) and 1:100 mouse anti-FANCI (sc-271316; Santa Cruz Biotechnology) or anti-V5 (66007.1-Ig; ProteinTech) for 60 min at room temperature, or overnight at 4°C. Membranes were washed extensively with PBS-T, incubated with secondary infrared-labelled antibodies (LI-COR) for 90 min at room temperature and then

washed extensively again with PBS-T. Bands were visualized on an Odyssey CLx (LI-COR) using the 700- or 800-nm channel.

## Ubiquitination assays

Ubiquitination of isolated FANCI was performed using His$_6$-FANCI or His$_6$-FANCI$^{R1285Q}$ (2 μM) as substrate, in the presence or absence of single-stranded DNA (ss64; 4 μM; Appendix Table S1). Reactions were conducted using His$_6$-Uba1 (50 nM), His$_6$-ubiquitin (5 μM), Ube2Tv4 (2 μM) and FANCL$^{109-375}$ (2 μM) in a final reaction buffer of 49 mM Tris pH 8.0, 120 mM NaCl, 5% glycerol, 1 mM DTT, 2.5 mM MgCl$_2$ and 2.5 mM ATP. Ubiquitination of ID2 or I$^{R1285Q}$D2 was performed in the presence of double-stranded DNA (ds50; Appendix Table S1). Reactions were prepared by dilution of all components into E3 reaction buffer (50 mM Tris pH 8.0, 75 mM NaCl, 5% glycerol, 5 mM MgCl$_2$ and 2.5 mM ATP) to yield final concentrations of 4 μM FANCI (His$_6$-FANCI or His$_6$-FANCI$^{R1285Q}$), 4 μM His$_6$-3C-FANCD2, 16 μM ds50, 4 μM Ube2Tv4, 4 μM FANCL$^{109-375}$, 100 nM His$_6$-Uba1 and 8 μM ubiquitin. All reactions were performed at room temperature and stopped at indicated time-points by addition of reducing LDS buffer (1× final concentration).

## Deubiquitination assays

The ID2$_{Ub}$-DNA and I$_{Ub}$D2$_{Ub}$-DNA substrates were reconstituted by mixing appropriate amount of D2$_{Ub}$, His$_6$-TEV-V5-FANCI or His$_6$-TEV-V5-I$_{Ub}$ and DNA (ds50; Appendix Table S1) to form a I/I$_{Ub}$:D2$_{Ub}$:DNA molar ratio of 1:1:4. D2$_{Ub}$-DNA substrate was prepared in the same way, but with GF buffer substituting I/I$_{Ub}$. The substrates were diluted in DUB buffer (50 mM Tris pH 8.0, 75 mM NaCl, 2 mM DTT, 5% glycerol) on ice to a concentration of 2 μM and incubated for at least 15 min. USP1-UAF1, USP7 or GST-USP2 were prepared at 2× concentrations in DUB buffer. To initiate reactions, 2× substrate was mixed 1:1 with 2× DUB in a 10 μl reaction volume and reactions were stopped at indicated time-points by addition of 10 μl of reducing 2× LDS buffer.

## DUB-step assays

His$_6$-TEV-V5-FANCI (4 μM) was ubiquitinated using Ube2Tv4 (4 μM), FANCL$^{109-375}$ (4 μM), His$_6$-Uba1 (100 nM), wild-type or mutant ubiquitin (8 μM) and DNA (ds50; 16 μM). All reaction components (apart from ubiquitin) were first diluted using E3 reaction buffer (50 mM Tris pH 8.0, 75 mM NaCl, 5% glycerol, 5 mM MgCl$_2$ and 2.5 mM ATP) and reactions subsequently initiated with addition of ubiquitin at room temperature. Reactions were arrested at indicated time-points by addition of 5 U/ml apyrase (New England Biolabs) and subsequent incubation on ice for 5 min. Ubiquitinated FANCI was subsequently mixed in a 1:1 ratio with purified D2$_{Ub}$ (2 μM ID2, 8 μM DNA) and incubated for a further 15 min on ice. The reconstituted complexes, D2$_{Ub}$, I$_{Ub}$-DNA or I$_{Ub}$D2$_{Ub}$-DNA, were then subject to deubiquitination by USP1-UAF1 (final concentrations 100 nM USP1-UAF1, 1 μM substrate) in 10 μl reaction volumes for an indicated amount of time at room temperature. Reactions were stopped by addition of reducing LDS buffer (1× final concentration). Deubiquitination progress was illustrated with a fitted one-phase decay model (GraphPad Prism).

## Electro-mobility shift assays

Indicated FANCI and FANCD2 protein constructs were pooled at equimolar concentrations and serially diluted in GF buffer. Each protein dilution was then mixed with EMSA reaction buffer for final concentrations of 2 nM labelled DNA (ds32$^F$; Appendix Table S1), 16 mM Tris–HCl, pH 8, 150 mM NaCl, 4.4% glycerol, 0.07 mg/ml BSA, 7 mM DTT, 4 mM EDTA and ID2 concentrations ranging from 4 to 512 nM. For assays testing the effect of salt concentration on binding, reconstituted ID2 and ID2$_{Ub}$ protein complexes (derived from His$_6$-TEV-V5-FANCI and either FLAG-FANCD2 or ubiquitinated FLAG-FANCD2) were diluted in 20 mM Tris pH 8, 200 mM NaCl, 5% glycerol and 7 mM DTT. Each protein dilution was mixed with EMSA reaction buffer to yield aforementioned final concentrations but with final NaCl being either 100 mM or 200 mM. Reactions (18 μl final) occurred on ice for 20 min, before addition of 2 μl of 10× Orange dye. 10 μl was then loaded on 4% polyacrylamide 0.5× TBE gels, that had been pre-run in 0.5× TBE buffer, and electrophoresis occurred at 135 V for 40–45 min. The gels were subsequently scanned in LI-COR imaging system (Odyssey CLx) using the 700-nm laser. Percentage of DNA bound was determined by the ratio of protein-bound signal to total DNA signal per lane.

## Protein induced fluorescence enhancement

For measurement of DNA binding, aliquots of FANCI (His$_6$-FANCI or His$_6$-TEV-V5-FANCI), FANCD2 (His$_6$-FANCD2 or His$_6$-3C-FANCD2) or the ubiquitinated versions in GF buffer were thawed on ice and then mixed at a molar ratio of 1:1 to form each different ID2 complex. Samples were then exchanged into Fluorescence Buffer (20 mM Tris pH 8.0, 150 mM NaCl, 5% glycerol, 0.47 mg/ml BSA, 1 mM DTT) by fivefold dilution with 20 mM Tris pH 8.0, 87.5 mM NaCl, 5% glycerol and 0.59 mg/ml BSA. Twofold serial dilutions of exchanged protein were set up in PCR tubes with Fluorescence Buffer. Labelled DNA (ds32$^F$), which was also diluted into Fluorescence Buffer, was mixed with each serial protein dilution to yield a final dsDNA concentration of 125 nM. For measurements testing the effect of salt, labelled DNA (ds32$^F$) was diluted 200-fold into 20 mM Tris pH 8.0, 50 mM NaCl, 5% glycerol, 0.47 mg/ml BSA, 1 mM DTT, or 20 mM Tris pH 8.0, 250 mM NaCl, 5% glycerol, 0.47 mg/ml BSA, 1 mM DTT, resulting in assay NaCl concentrations of 100 and 200 mM, respectively.

For measurement of FANCI-FANCD2 interaction, aliquots of FANCD2 (deriving from His$_6$-3C-FANCD2 in which the His$_6$-tag was cleaved by 3C protease) and His$_6$-FANCI or His$_6$-FANCI R1285Q were thawed on ice. Samples were then exchanged into Fluorescence Buffer by fivefold dilution with 20 mM Tris pH 8.0, 87.5 mM NaCl, 5% glycerol and 0.59 mg/ml BSA. Twofold serial dilutions of exchanged FANCD2 were set up in PCR tubes with Fluorescence Buffer. 120 nM exchanged His$_6$-FANCI was labelled with 50 nM RED-tris-NTA dye (NanoTemper) in Fluorescence Buffer and added to the serial dilution to yield a final concentration of 60 nM His$_6$-FANCI and 25 nM dye.

For measurement of DNA binding of FANCI ubiquitin mutants in complex with D2$_{Ub}$, His$_6$-TEV-V5-FANCI was ubiquitinated as per the DUB-step assays (using a reaction buffer of 10 mM Tris pH 8.5, 75 mM NaCl, 5% glycerol, 2.5 mM MgCl$_2$ and 2.5 mM ATP) and terminated by addition of 5 U/ml apyrase. 15 μl of this reaction mix

was added to 5 μl of His$_6$-FANCD2$_{Ub}$ (or GF buffer for the no D2$_{Ub}$ control) to yield a final concentration of 3 μM I$_{Ub}$D2$_{Ub}$ (or I$_{Ub}$) and a NaCl concentration of ~ 150 mM. Twofold serial dilutions of this complex were set up in matched buffer (including all the reaction components minus ubiquitin), and each was mixed in 1:1 ratio with labelled DNA (ds32$^F$; 250 nM) in Fluorescence Buffer to yield a final dsDNA concentration of 125 nM.

Prior to fluorescence measurement, samples were briefly centrifuged and then transferred into premium capillaries (NanoTemper Technologies). Measurements were performed at 22°C on a Monolith NT.115 instrument (NanoTemper Technologies) using the red channel. Laser power was set to 20 and 40% for DNA binding and FANCD2 binding, respectively.

### Fitting of binding data

Binding affinities and associated uncertainties were determined with non-linear regression in GraphPad Prism by fitting a one-site binding model:

$$Y = F_0 + (F_1 - F_0) \frac{[A_T] + [B_T] + K_d - \sqrt{([A_T] + [B_T] + K_d)^2 - 4[A_T][B_T]}}{2[A_T]}$$

where $Y$ is either the fluorescence change (in the case of PIFE) or the percentage DNA binding (in the case of EMSAs), $F_0$ is baseline, $F_1$ is the plateau, $[A_T]$ is the constant concentration of the fluorescent binding molecule, and $[B_T]$ is the varying concentration of the other binding molecule/complex. For EMSAs, the baseline was constrained to 0 and the plateau was shared between analysed datasets. For PIFE, fitted curves of Fig 1A were constrained to have a shared amplitude ($F_1$–$F_0$), which was fixed for fitted curves of Figs 3E and EV5. Fitted curves of Fig EV1A were constrained to have a shared amplitude. The baseline was subtracted in the plotted data and curves.

### CryoEM sample preparation

FANCD2$_{Ub}$ and His$_6$-TEV-V5-FANCI or His$_6$-TEV-V5-FANCI$_{Ub}$ aliquots were thawed and mixed at a molar ratio of 1:1 and then exchanged into EM Buffer (20 mM Tris pH 8.0, 100 mM NaCl, 2 mM DTT) using a Zeba™ Spin 7K MWCO desalting column. For I$_{Ub}$D2$_{Ub}$-dsDNA, ds32 (Appendix Table S1) was then added at a molar ratio of 1:1:1 I$_{Ub}$:D2$_{Ub}$:DNA. The samples were diluted to 3 μM (I$_{Ub}$D2$_{Ub}$-dsDNA) or 1.5 μM (ID2$_{Ub}$) of complex. 3.5 μl of sample was applied to glow discharged grids (C-Flat 2/2 or Quantifoil 2/2), blotted for 2.5–3.5 s and cryo-cooled in liquid ethane using a Vitrobot operating at ~ 95% humidity at 4.5°C.

### CryoEM data collection and image processing

For I$_{Ub}$D2$_{Ub}$-dsDNA, 1,846 movies were collected on a Titan Krios (Thermo Fisher Scientific) equipped with a Falcon III detector operating in counting mode using EPU software (Thermo Fisher Scientific). Each movie was 60 frames and motion-corrected with dose-weighting using MotionCor2-1.1.0. For the ID2$^{Ub}$ sample, 1,146 and 916 movies were collected in two sessions on a 300 kV CRYOARM (JEOL) equipped with a DE64 detector

operating in counting mode using SerialEM [40]. Each movie was 39 frames and additional gain correction applied via relion_estimate_gain. For both datasets, CTF correction was performed using gCTF [41]. Further data collection details are provided in Appendix Table S2.

Subsequence processing was performed using RELION 3.0 or 3.1 [42]. For the I$_{Ub}$D2$_{Ub}$-dsDNA dataset, approximately 5,000 particles were manually picked and used to generate reference-free 2D class averages. Selected class averages were then used as templates to auto-pick particles. 350,878 particles were extracted, and five rounds of reference-free 2D class averaging were used to remove poor particles. An initial model was generated using stochastic gradient descent followed by one round of 3D classification with six classes. Particles from the highest estimated resolution class (12,710) were then used in 3D refinement. A mask was then generated, and post-processing performed. For the ID2$_{Ub}$ dataset particles, 7,810 particles were manually picked, extracted at 4.784 Å/px, and one round of reference-free 2D classification was performed to remove poor particles. An initial model was generated using stochastic gradient descent followed by one round of 3D classification with two classes. Particles from the best class (4,404) were then used in 3D refinement.

The mouse ID2 structure (PDB: 3S4W) [18] was fit as a rigid body into the final I$_{Ub}$D2$_{Ub}$-dsDNA map using UCSF Chimera [43]. Flexible fitting of Cα atoms was subsequently performed using iMODFIT [31] incorporating secondary structure constraints and using data to 17 Å. The fitted ID2 structure was then used to generate a map at 12 Å, which was subtracted from the experimental map using UCSF Chimera to identify the difference in density.

## Data availability

The data produced in this study are available as follows:

- CryoEM reconstruction (ID2$_{Ub}$): EMDB EMD-10843 (https://www.ebi.ac.uk/pdbe/entry/emdb/EMD-10843)
- CryoEM reconstruction (I$_{Ub}$D2$_{Ub}$-dsDNA): EMDB EMD-10844 (https://www.ebi.ac.uk/pdbe/entry/emdb/EMD-10844)
- PIFE raw and baseline subtracted data: Source Data associated with this manuscript.

**Expanded View** for this article is available online.

## Acknowledgements

We thank past and current members of the Walden laboratory for experimental suggestions, comments on the manuscript and their support. All constructs are available on request from the MRC Protein Phosphorylation and Ubiquitylation Unit reagents Web page (http://mrcppureagents.dundee.ac.uk) or from the corresponding author. We acknowledge Diamond Light Source for time on Krios IV under proposal EM16637 and thank Dr. Daniel Clare for collecting data. We acknowledge the Scottish Centre for Macromolecular Imaging (SCMI) for access to cryoEM instrumentation, funded by the MRC (MC_PC_17135) and SFC (H17007). We thank Ms. June Southall (University of Glasgow) for assistance for the PIFE experiments. This work was supported by the EMBO Young Investigator Programme to H.W. and the European Research Council (ERC-2015-CoG-681582 ICLUb) consolidator grant to H.W.

## Author contributions

MLR, CA, KL and HW conceived this work; MLR, KL, CA and VKC purified proteins; MLR, KL and CA designed and executed experiments; MLR performed PIFE and ubiquitination assays; KL performed EMSA experiments; CA and MLR performed DUB-associated assays; MC prepared cryoEM grids; JS collected CRYOARM cryoEM data; MLR performed cryoEM data analysis with guidance from LS; KL performed quantification and statistical analyses; KL made the figures with input from MLR; MLR, KL and CA wrote the manuscript; MLR, KL and CA edited the manuscript with contributions from all other authors; and HW secured funding and supervised the project.

## Conflict of interest

The authors declare that they have no conflict of interest.

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
