## [Review Process File · EMBO Reports]

Differential functions of FANCI and FANCD2 ubiquitination stabilize ID2 complex on DNA

Helen Walden, Martin Rennie, Kimon Lemonidis, Connor Arkinson, Viduth Chaugule, Mairi Clarke, James Streetley, and Laura Spagnolo

DOI: [10.15252/embr.202050133](https://doi.org/10.15252/embr.202050133)

Corresponding author(s): Helen Walden (Helen.Walden@glasgow.ac.uk)

Review Timeline:

Submission Date:	31st Jan 20
Editorial Decision:	11th Mar 20
Additional Correspondence:	6th Apr 20
Revision Received:	9th Apr 20
Editorial Decision:	13th May 20
Revision Received:	15th May 20
Accepted:	15th May 20

Transaction Report:

Dear Helen,

Thank you for the submission of your manuscript to EMBO reports. I apologize again for the delay in handling your manuscript. As I already informed you, we have meanwhile received two reports on it but are still waiting for the third report from the structural biology expert. Since the two reports we have already received are in agreement and since both referees support a revision for EMBO reports, we have decided to invite you to begin revising your study along the lines suggested by the referees. As soon as we will receive the final report on your manuscript, we will forward it to you as well.

As you will see, the referees acknowledge that the findings are potentially interesting. However, referees 1 and 2 also point out several technical concerns and have a number of suggestions for how the study should be strengthened. Referee 1 considers the prediction and testing of FANCI and FANCD2 mutants in a cellular context important. Experiments along these lines should be added if not too time-consuming. Referee 1 further recommends testing for the effect of residual DNA on ubiquitination and DNA binding. Referee 2 suggests testing the DNA affinity of the FANCIUb-FANCD2 complex. Upon further discussion of this point with referee 2, s/he indicated that the experiment should be straightforward as you have all materials at hand but also stated that "...since this is such an obvious experiment the authors might have a good reason why this did not or will not work. In that case I would be fine with this as long as they provide this reasoning". Thus either address this point experimentally or discuss it in the text.

Given these constructive comments, we would like to invite you to revise your manuscript with the understanding that the referee concerns (as detailed above and in their reports) must be fully addressed and their suggestions taken on board. Please address all referee concerns in a complete point-by-point response. Acceptance of the manuscript will depend on a positive outcome of a second round of review. It is EMBO reports policy to allow a single round of revision only and acceptance or rejection of the manuscript will therefore depend on the completeness of your responses included in the next, final version of the manuscript.

Revised manuscripts should be submitted within three months of a request for revision; they will otherwise be treated as new submissions. Please contact us if a 3-months time frame is not sufficient for the revisions so that we can discuss the revisions further.

- 1) A data availability section is missing.
- 2) Your manuscript contains error bars based on $n=2$. Please use scatter blots showing the individual datapoints in these cases. The use of statistical tests needs to be justified.

- 1) a .docx formatted version of the manuscript text (including legends for main figures, EV figures

and tables). Please make sure that the changes are highlighted to be clearly visible.

2) You have currently five figures and your study will thus be published in our Reports section. This requires that you combine the Results and Discussion section and please keep our character limit of 25,000 (+/- 2,000) characters in mind (excluding references and materials and methods).

3) individual production quality figure files as .eps, .tif, .jpg (one file per figure).

Please download our Figure Preparation Guidelines (figure preparation pdf) from our Author Guidelines pages

<https://www.embopress.org/page/journal/14693178/authorguide> for more info on how to prepare your figures.

5) a complete author checklist, which you can download from our author guidelines (). Please insert information in the checklist that is also reflected in the manuscript. The completed author checklist will also be part of the RPF.

6) Please note that all corresponding authors are required to supply an ORCID ID for their name upon submission of a revised manuscript (). Please find instructions on how to link your ORCID ID to your account in our manuscript tracking system in our Author guidelines

()

7) We replaced Supplementary Information with Expanded View (EV) Figures and Tables that are collapsible/expandable online. A maximum of 5 EV Figures can be typeset. EV Figures should be cited as 'Figure EV1, Figure EV2" etc... in the text and their respective legends should be included in the main text after the legends of regular figures.

- For the figures that you do NOT wish to display as Expanded View figures, they should be bundled together with their legends in a single PDF file called *Appendix*, which should start with a short Table of Content (including page numbers). Appendix figures should be referred to in the main text as: "Appendix Figure S1, Appendix Figure S2" etc. See detailed instructions regarding expanded view here:

8) Before submitting your revision, primary datasets (and computer code, where appropriate) produced in this study need to be deposited in an appropriate public database (see < <https://www.embopress.org/page/journal/14693178/authorguide#dataavailability>>). Specifically, we would kindly ask you to provide public access to the cryo-EM data.

The accession numbers and database should be listed in a formal "Data Availability " section (placed after Materials & Method) that follows the model below (see also < <https://www.embopress.org/page/journal/14693178/authorguide#dataavailability>>). Please note that the Data Availability Section is restricted to new primary data that are part of this study.

Data availability

9) We would also encourage you to include the source data for figure panels that show essential data. Numerical data should be provided as individual .xls or .csv files (including a tab describing the data). For blots or microscopy, uncropped images should be submitted (using a zip archive if multiple images need to be supplied for one panel). Additional information on source data and instruction on how to label the files are available .

10) Our journal encourages inclusion of *data citations in the reference list* to directly cite datasets that were re-used and obtained from public databases. Data citations in the article text are distinct from normal bibliographical citations and should directly link to the database records from which the data can be accessed. In the main text, data citations are formatted as follows: "Data ref: Smith et al, 2001" or "Data ref: NCBI Sequence Read Archive PRJNA342805, 2017". In the Reference list, data citations must be labeled with "[DATASET]". A data reference must provide the database name, accession number/identifiers and a resolvable link to the landing page from which the data can be accessed at the end of the reference. Further instructions are available at .

11) Regarding data quantification:

- Please ensure to specify the name of the statistical test used to generate error bars and P values, the number (n) of independent experiments underlying each data point (not replicate measures of one sample), and the test used to calculate p-values in each figure legend. Discussion of statistical methodology can be reported in the materials and methods section, but figure legends should contain a basic description of n, P and the test applied.

IMPORTANT: Please note that error bars and statistical comparisons may only be applied to data obtained from at least three independent biological replicates. If the data rely on a smaller number of replicates, scatter blots showing individual data points are recommended.

- Graphs must include a description of the bars and the error bars (s.d., s.e.m.).
- Please also include scale bars in all microscopy images.

12) As part of the EMBO publication's Transparent Editorial Process, EMBO reports publishes online a Review Process File to accompany accepted manuscripts. This File will be published in conjunction with your paper and will include the referee reports, your point-by-point response and all pertinent correspondence relating to the manuscript.

You are able to opt out of this by letting the editorial office know (emboreports@embo.org). If you

do opt out, the Review Process File link will point to the following statement: "No Review Process File is available with this article, as the authors have chosen not to make the review process public in this case."

I look forward to seeing a revised version of your manuscript when it is ready. Please let me know if you have questions or comments regarding the revision.

Kind regards,
Martina

Martina Rembold, PhD
Editor
EMBO reports

Referee #1:

The paper by Walden and colleagues describes a function of FANCD2 and FANCI in stabilization of protein that depends on ubiquitination and DNA binding. Overall these are interesting data but would benefit by attention to a few additional details as listed below. In addition, some cell biology correlates would be helpful to assure confidence. For example, can any prediction be made and subsequent FANCI or FANCD2 mutants expressed that would have functional effects in cells? Technically, ubiquitination and deubiquitination depend on DNA binding, and the retained DNA may have some effect on assessment of downstream effects, such as DNA affinity. A benzonase treatment should be inserted to account for this to remove DNA and then proceed to DNA binding experiments to assure the fidelity of the results presented. I could not find any statement in the text to this effect. In addition, it may not be possible to assess the behavior of isolated D2 as in many hands appears to form aggregates.

From an interpretation point of view, one can argue that the cryoEM suggests that the ID2-ub complex closes against the DNA which is present during the ub reaction. This would seem to be an alternative explanation to that suggested by the DNA binding reactions in Fig 1.

For the de ub assays, again the question of DNA being present or not is important. Recent reports are suggestive that it is, and this the variable of DNA binding should be assessed. While DNA was added, it is unclear if as a control DNA was omitted. Again, benzonase may have to be used as a pre treatment to ensure DNA is absent prior to addition/omission of DNA for the reaction.

Minor

The manuscript needs editing for awkward constructs, grammar, etc; see abstract

Referee #2:

This manuscript by Rennie et al. describes the biochemical characterization of the role of ubiquitination of the FANCI-FANCD2 (ID2) complex. While this posttranslational modification has long been known to activate the FA-pathway that repairs DNA interstrand crosslinks, the biochemical function of ubiquitination has remained a mystery. This has been in part due to the fact

that it has proven difficult to purify the ubiquitinated form of the ID2 complex. The Walden lab was one of the first labs to be able to do this, although other labs (e.g. Passmore, Deans and Pavletich) are also reporting on this. The authors purified the unmodified ID2 complex, the complex with site-specific ubiquitinated FANCD2, or ubiquitinated FANCD2 and FANCI, and determined their DNA binding affinities. They show that ubiquitination enhances the affinity up to ten-fold. They then performed cryoEM on the FANCI-FANCD2Ub and the FANCIUb-FANCD2Ub complexes. While the resolution of the cryoEM maps is low, aided by modeling they show that ubiquitination (on one or both subunits) induces a structural rearrangement to a closed complex encompassing DNA. They hypothesize that this new conformation may be important for ID2 ubiquitination, and that a patient mutation (R1285Q), that is positioned close to the new ID2Ub interface and may disrupt it, could affect modification. Subsequently they demonstrated that this FANCI mutant does not affect FANCI ubiquitination but reduces FANCD2 ubiquitination. Moreover, the affinity for DNA is not enhanced in the FANCI^{R1285Q}-FANCD2Ub as much as in the FANCI-FANCD2Ub compared to the non-ubiquitinated complexes indicating this mutation prevents the formation of the new closed conformation. Finally, since FANCI ubiquitination does not seem to be required for enhanced DNA binding they investigate its role in deubiquitination by USP1-UAF1. As was indicated previously, they show that deubiquitination is inhibited in the FANCIUb-FANCD2Ub complex compared to the FANCI-FANCD2Ub complex. Using ubiquitin mutants that are known to affect interaction with other proteins they show that specific ubiquitin residues are required to promote the resistance to deubiquitination, likely through interaction with FANCD2.

This is an elegant study containing a series of well executed experiments. The mechanism in which the ubiquitins on FANCI and FANCD2 have to different biochemical functions that lead to the same result, the stabilization of the ID2 complex on DNA, is very interesting and of major interest to the field. Although the findings in this study should be published, I have a few issues with the conclusions that should be addressed.

First, an obvious complex that is not studied is the FANCIUb-FANCD2 complex. Although the FANCD2 ubiquitination seems to precede FANCI ubiquitination in bulk assays, it is not unlikely that this complex can also form. Would this complex induce the same structural rearrangements and enhance the affinity for DNA? If that is the case the role of the ubiquitination of FANCI and FANCD2 would not be as separate as the authors suggest. This needs to be resolved.

Second, the authors state that the closed conformation is important for ubiquitination of FANCD2 by exposing the target lysine that is buried in the open conformation. If this is the case, this would also be true for FANCI. However, FANCI ubiquitination does not seem to be affected by the R1285Q mutations (Fig. 3d) that supposedly inhibits the closed conformation. The authors should explain why.

Finally, there is a 5 to 20-fold difference in ID2 - DNA affinities between the EMSA and PIFE measurements (Fig. 1 and 3). The authors should explain why this is the case.

Dear Helen,

we have now received the report from the structural biology expert, referee #3.

As you will see, also this referee is very positive and supports publication of your manuscript. Referee #3 suggested to test the conformational change and the FANCI R1285Q mutation using SAXS analysis. Given that this report was delivered with a delay, given the current situation, given that several related manuscripts have already been published and given that referee 3 indicated that this experiment is a suggestion to strengthen your otherwise convincing dataset, I suggest to address only the other concerns from this referee and maybe discuss potential further experiments in the text.

I am looking forward to receiving your revised manuscript.

Kind regards,
Martina

Martina Rembold, PhD
Editor
EMBO reports

Referee #3

Rennie et al report the effect of mono-ubiquitination of FANCI (I) and FANCD2 (D2) in the ID2 complex results in a conformation change that forms a novel C-terminal interface between I and D2. This interface enhances overall DNA binding affinity of the complex which may play a regulatory role in localizing the complex on DNA during DNA damage repair. Further, ubiquitination of FANCI protects the ubiquitination on FANCD2 against USP1-UAF1. Interestingly, the pathogenic mutation on FANCI R1285Q which is located on the C-terminal interface causes decrease in the DNA binding affinity.

ID2 complex is at the nexus of the interstrand crosslink (ICL) DNA damage repair involving Fanconi anemia pathway. The ubiquitination of ID2 is a key event that connects the recognition, initiation to the downstream repair processes of ICL damage. In this article, by using recombinant proteins with ubiquitin modification in cryogenic electron microscopy with modest resolution over 10 Å, author have shown that there is a novel C-ter interface formation upon ubiquitination of ID2. These observations are in agreement with the recent two high-resolution studies published from Paveltich (Nature 2020) and Passmore (NSMB 2020) groups. Further, their observation on enhanced DNA binding upon ubiquitination is also supported by one more contemporary article from Deans (Elife 2020) group. All the statements in this article are supported by their data from elegant biochemistry experiments with no major concerns. However, the should consider the following points.

1) The hypothesis is that the pathogenic mutation R1285Q in FANCI disrupts the C-ter interface formation which may cause the decreased DNA binding of the overall ID2 complex. Since authors have unique capability to make recombinant mutant FANCI and Ub-FANCD2, and the conformation change is significant for the C-ter interface formation, why not test this? They should be able to test this with solution-based techniques such as SAXS. This is not only a key data for the current paper, but also will make this study unique from other recently published similar papers. This is a suggestion only.

Minor points-

- 1) In figure 3F, although there is a fitting curve but no data points are given for ID2 WT
- 2) Although PIFE experiments seems to be done in replicates the error bars on data points in the figures are missing (Figure 1B, 3F, EV2B)
- 3) There are typos in the article- e.g. Figure 3 legend has R1286Q instead of R1285Q, in the abstract, interphase instead of interface...

As you will see, the referees acknowledge that the findings are potentially interesting. However, referees 1 and 2 also point out several technical concerns and have a number of suggestions for how the study should be strengthened. Referee 1 considers the prediction and testing of FANCI and FANCD2 mutants in a cellular context important. Experiments along these lines should be added if not too time-consuming. Referee 1 further recommends testing for the effect of residual DNA on ubiquitination and DNA binding. Referee 2 suggests testing the DNA affinity of the FANCIUb-FANCD2 complex. Upon further discussion of this point with referee 2, s/he indicated that the experiment should be straightforward as you have all materials at hand but also stated that "...since this is such an obvious experiment the authors might have a good reason why this did not or will not work. In that case I would be fine with this as long as they provide this reasoning". Thus either address this point experimentally or discuss it in the text.

Referee #1:

The paper by Walden and colleagues describes a function of FANCD2 and FANCI in stabilization of protein that depends on ubiquitination and DNA binding. Overall these are interesting data but would benefit by attention to a few additional details as listed below. In addition, some cell biology correlates would be helpful to assure confidence. For example, can any prediction be made and subsequent FANCI or FANCD2 mutants expressed that would have functional effects in cells?

We agree that cell-based experiments would be useful. We have thus requested cell lines for FANCI and FANCD2 knockouts to perform these, as we are not a cell biology lab. Unfortunately, due to the COVID-19 situation we are unable to give a timeframe for completing these experiments and therefore request you consider the manuscript without them. We have included some discussion on future cell-based experiments in the text.

Regarding the confidence on our results, three groups have now published work where each independently observes enhancement of ID2-DNA-binding due to FANCD2 ubiquitination.

Technically, ubiquitination and deubiquitination depend on DNA binding, and the retained DNA may have some effect on assessment of downstream effects, such as DNA affinity. A benzonase treatment should be inserted to account for this to remove DNA and then proceed to DNA binding experiments to assure the fidelity of the results presented. I could not find any statement in the text to this effect.

The reviewer correctly points out that DNA is required for ID2 ubiquitination *in vivo* and *in vitro*. However, in our lab we use a different approach to ubiquitinate FANCD2 and FANCI. We perform ubiquitination reactions with isolated FANCI or isolated FANCD2 and do not use DNA to enhance the reactions. Instead we use an engineered Ube2T (Arkinson *et al* 2018, Chaugule *et al* 2019, Chaugule *et al* 2020 – refs 25,26, and 32). We have added relevant words/sentences in our text to make it clear that we ubiquitinate FANCD2 and FANCI separately, and in the absence of DNA.

We also note that our purification of FANCI and FANCD2 is similar to other groups that have looked at binding to DNA and we perform a benzonase treatment during purification of FANCI and FANCD2 (following cell lysis). All protein material used had a 260/280 ratio <0.8 (generally around 0.6-0.65) suggesting these are essentially DNA-free. We have updated the text to describe this.

To further support this we show below control ubiquitination reactions of FANCI or FANCD2 in the presence or absence of benzonase and observe no difference (Figure R1), which we are happy include in the paper if the reviewer considers it to be necessary.

Figure R1. Ubiquitination of purified FANCI or FANCD2 was performed at 1 μ M substrate, 0.05 μ M Uba1, 1 μ M Ube2Tv4, 1 μ M FANCL¹⁰⁹⁻³⁷⁵, 8 μ M ubiquitin, 2.5 mM ATP, 2.5 mM MgCl₂. Ubiquitin and reaction components lacking ubiquitin were separately treated with 12 U/ μ L benzonase (Expedeon BaseMuncher) for 30 min at room temperature, and then mixed to initiate each reaction.

In addition, it may not be possible to assess the behavior of isolated D2 as in many hands appears to form aggregates.

During our purification of D2_{Ub} minimal aggregation is observed (Chaugule *et al* 2019 – ref 26). We would argue that our observation that FANCI is required for enhancement of DNA binding upon FANCD2 ubiquitination remains valid whether or not D2_{Ub} aggregates in the absence of FANCI.

From an interpretation point of view, one can argue that the cryoEM suggests that the ID2-ub complex closes against the DNA which is present during the ub reaction. This would seem to be an alternative explanation to that suggested by the DNA binding reactions in Fig 1.

We believe that the cryoEM data is consistent with the DNA binding reactions, whereby the closed conformation is accessed when ID2 binds DNA and is stabilized via ubiquitination of FANCD2. We note that we did not include DNA during FANCI/FANCD2 ubiquitination (as mentioned above) and hence all FANCI/FANCI_{Ub} and FANCD2/FANCD2_{Ub} samples (used for reconstitution of ubiquitinated ID2_{Ub}, I_{Ub}D2_{Ub} or I_{Ub}D2_{Ub}-DNA complexes) were essentially free of DNA. DNA was added only for reconstitution of the I_{Ub}D2_{Ub}-DNA complex assessed by cryoEM.

For the de ub assays, again the question of DNA being present or not is important. Recent reports are suggestive that it is, and this the variable of DNA binding should be assessed. While DNA was added, it is unclear if as a control DNA was omitted. Again, benzonase may have to be used as a pre treatment to ensure DNA is absent prior to addition/omission of DNA for the reaction.

We have previously shown the effect of removing DNA from ID2_{Ub}-DNA and I_{Ub}D2_{Ub}-DNA complexes on FANCD2/FANCI deubiquitination (Arkinson *et al* 2018 – ref 32). There we showed that benzonase treatment made I_{Ub}D2_{Ub}-DNA complexes more susceptible to USP1-UAF1 deubiquitination.

Minor

The manuscript needs editing for awkward constructs, grammar, etc; see abstract
We thank the reviewer for highlighting this. We have carefully been through the text and re-worded the appropriate sentences.

Referee #2:

This manuscript by Rennie et al. describes the biochemical characterization of the role of ubiquitination of the FANCI-FANCD2 (ID2) complex. While this posttranslational modification has long been known to activate the FA-pathway that repairs DNA interstrand crosslinks, the biochemical function of ubiquitination has remained a mystery. This has been in part due to the fact that it has proven difficult to purify the ubiquitinated form of the ID2 complex. The Walden lab was one of the first labs to be able to do this, although other labs (e.g. Passmore, Deans and Pavletich) are also reporting on this. The authors purified the unmodified ID2 complex, the complex with site-specific ubiquitinated FANCD2, or ubiquitinated FANCD2 and FANCI, and determined their DNA binding affinities. They show that ubiquitination enhances the affinity up to ten-fold. They then performed cryoEM on the FANCI-FANCD2Ub and the FANCIUb-FANCD2Ub complexes. While the resolution of the cryoEM maps is low, aided by modeling they show that ubiquitination (on one or both subunits) induces a structural rearrangement to a closed complex encompassing DNA. They hypothesize that this new conformation may be important for ID2 ubiquitination, and that a patient mutation (R1285Q), that is positioned close to the new ID2Ub interface and may disrupt it, could affect modification. Subsequently they demonstrated that this FANCI mutant does not affect FANCI ubiquitination but reduces FANCD2 ubiquitination. Moreover, the affinity for DNA is not enhanced in the FANCI^{R1285Q}-FANCD2Ub as much as in the FANCI-FANCD2Ub compared to the non-ubiquitinated complexes indicating this mutation prevents the formation of the new closed conformation. Finally, since FANCI ubiquitination does not seem to be required for enhanced DNA binding they investigate its role in deubiquitination by USP1-UAF1. As was indicated previously, they show that deubiquitination is inhibited in the FANCIUb-FANCD2Ub complex compared to the FANCI-FANCD2Ub complex. Using ubiquitin mutants that are known to affect interaction with other proteins they show that specific ubiquitin residues are required to promote the resistance to deubiquitination, likely through interaction with FANCD2.

This is an elegant study containing a series of well executed experiments. The mechanism in which the ubiquitins on FANCI and FANCD2 have to different biochemical functions that lead to the same result, the stabilization of the ID2 complex on DNA, is very interesting and of major interest to the field. Although the findings in this study should be published, I have a few issues with the conclusions that should be addressed.

First, an obvious complex that is not studied is the FANCIUb-FANCD2 complex. Although the FANCD2 ubiquitination seems to precede FANCI ubiquitination in bulk assays, it is not unlikely that this complex can also form. Would this complex induce the same structural rearrangements and enhance the affinity for DNA? If that is the case the role of the ubiquitination of FANCI and FANCD2 would not be as separate as the authors suggest. This needs to be resolved.

We agree that the I_{Ub}D2 state is important to test for completeness sake and also because our reconstitution setup readily allows this. We have included DNA binding assays with I_{Ub}D2 and find an intermediate DNA-affinity between ID2 and ID2_{Ub}

(Figure EV5). Given that the unmodified ID2 interface partially blocks this site, we expect some structural changes, which would result in the partially enhanced affinity. We have updated the text to incorporate this.

Second, the authors state that the closed conformation is important for ubiquitination of FANCD2 by exposing the target lysine that is buried in the open conformation. If this is the case, this would also be true for FANCI. However, FANCI ubiquitination does not seem to be affected by the R1285Q mutations (Fig. 3d) that supposedly inhibits the closed confirmation. The authors should explain why.

Indeed we did not observe any major effect of FANCI R1285Q mutant on FANCI ubiquitination (within an ID2 complex) in our assays. We do not currently know why this occurs. However, we think that this may be due to an enhanced FANCI^{R1285Q} ubiquitination (relative to FANCI^{WT}) when FANCD2 is ubiquitinated (see preliminary data in Figure R2).

Figure R2. ID2_{Ub} complexes were reconstituted using either wild-type (I^{WT}) or mutant (I^{R1285Q}) His₆-FANCI, and FANCI ubiquitination progress was observed by western blotting using a FANCI antibody. Ubiquitination was performed at 1 μM substrate, 0.05 μM Uba1, 1 μM Ube2Tv4, 1 μM FANCL¹⁰⁹⁻³⁷⁵, 8 μM ubiquitin, 2.5 mM ATP, 2.5 mM MgCl₂.

Nevertheless, we think the minimal FANCI ubiquitination we observe in Fig 3D is not physiologically relevant, since no FANCI/FANCD2 ubiquitination is observed *in vivo* for this mutation, and that some other mechanism is at play for FANCI ubiquitination not captured in our assay setup.

Moreover recent *in vitro* studies have shown that when FANCI is mutated to R1285Q, FA-core complex ubiquitinates FANCI/FANCD2 slower, whereas USP1-UAF1 deubiquitinate FANCI/FANCD2 faster. We thus replaced the relevant section in our manuscript with the following sentences:

“Nevertheless, the FANCI mutation resulted in an apparent reduction in FANCD2 ubiquitination in the ID2 complex (Figure 3D), consistent with previous results [20,21,28]. Under our assay conditions we did not detect a significant change in FANCI ubiquitination in the ID2 complex due to the mutation. Nevertheless, the FANCI R1285Q mutation was recently shown to result, not only in a reduction of FA-core catalysed FANCI and FANCD2 ubiquitination within an ID2 complex [14], but also in faster deubiquitination of the ubiquitinated complex [28]. This slower ubiquitination and faster deubiquitination may explain the nearly complete absence of ubiquitinated FANCD2/FANCI seen in cells having the FANCI R1285Q mutation [7].”

Finally, there is a 5 to 20-fold difference in ID2 - DNA affinities between the EMSA and PIFE measurements (Fig. 1 and 3). The authors should explain why this is the case.

We indeed observed large differences in DNA-affinities measured with these two techniques. Unlike PIFE, EMSAs operate under non-equilibrium conditions and the

low ionic strength of the electrophoresis buffer likely increases the apparent affinity. To support this, we have added PIFE and EMSA assays at varied salt concentrations of the sample buffer and find that affinities determined by EMSAs are relatively unaffected by the sample salt concentration, unlike PIFE (Figure EV1). As protein-DNA interactions are generally dependent on ionic strength, we have focussed on PIFE for determining and comparing dissociation constants. We have updated the text to describe this. Additionally, we removed relevant graphs showing binding affinities/dissociation constants determined by EMSA (Fig 1 & Fig 3), since these may be confusing to the reader and less relevant (than PIFE data) for the salt conditions under which the experiments were conducted.

Referee #3

Rennie et al report the effect of mono-ubiquitination of FANCI (I) and FANCD2 (D2) in the ID2 complex results in a conformation change that forms a novel C-terminal interface between I and D2. This interface enhances overall DNA binding affinity of the complex which may play a regulatory role in localizing the complex on DNA during DNA damage repair. Further, ubiquitination of FANCI protects the ubiquitination on FANCD2 against USP1-UAF1. Interestingly, the pathogenic mutation on FANCI R1285Q which is located on the C-terminal interface causes decrease in the DNA binding affinity.

ID2 complex is at the nexus of the interstrand crosslink (ICL) DNA damage repair involving Fanconi anemia pathway. The ubiquitination of ID2 is a key event that connects the recognition, initiation to the downstream repair processes of ICL damage. In this article, by using recombinant proteins with ubiquitin modification in cryogenic electron microscopy with modest resolution over 10 Å, authors have shown that there is a novel C-ter interface formation upon ubiquitination of ID2. These observations are in agreement with the recent two high-resolution studies published from Paveltich (Nature 2020) and Passmore (NSMB 2020) groups. Further, their observation on enhanced DNA binding upon ubiquitination is also supported by one more contemporary article from Deans (Elife 2020) group. All the statements in this article are supported by their data from elegant biochemistry experiments with no major concerns. However, they should consider the following points.

1) The hypothesis is that the pathogenic mutation R1285Q in FANCI disrupts the C-ter interface formation which may cause the decreased DNA binding of the overall ID2 complex. Since authors have unique capability to make recombinant mutant FANCI and Ub-FANCD2, and the conformation change is significant for the C-ter interface formation, why not test this? They should be able to test this with solution-based techniques such as SAXS. This is not only a key data for the current paper, but also will make this study unique from other recently published similar papers. This is a suggestion only.

We agree further solution based evidence for conformational changes would be useful. Unfortunately we do not have the resources available at the present stage due to the COVID-19 however we have updated the text to discuss further potential experiments.

Minor points-

1) In figure 3F, although there is a fitting curve but no data points are given for ID2 WT

Curves from WT are transferred from Figure 1A, which were performed identically to the mutant. We decided not to include the WT data points here to avoid data

duplication. We have edited both the figure and corresponding figure legend to highlight this further.

2) Although PIFE experiments seems to be done in replicates the error bars on data points in the figures are missing (Figure 1B, 3F, EV2B)

Most PIFE replicates were performed with different starting ID2 concentrations for two-fold serial dilutions. All data points from different experiments are plotted in the graphs and used together for Kd determination. We have now included the raw data for PIFE with the manuscript.

3) There are typos in the article- e.g. Figure 3 legend has R1286Q instead of R1285Q, in the abstract, interphase instead of interface...

We thank the reviewer for pointing these out. We have carefully been through the text and corrected these and other typos.

Dear Helen,

Thank you for the submission of your revised manuscript to EMBO reports. The revision was evaluated by referee 1 and 2 and we have now received their reports (copied below).

Referee 2 is very positive about the study and supports publication. Referee 1 also considered the revision adequate but noted the absence of cell-based experiments. Given that we had agreed that these experiments are not essential for publication in EMBO Reports at this point, I am happy to inform you that we can proceed with publication after a few editorial things have been addressed as follows:

- Please provide up to five keywords.
- Please add a Conflict of Interest statement to the article.
- Please update the callout to "Table S1" on page 15 to "Appendix Table S1"
- Could a scale bar be inserted in Fig. EV2?
- I attach to this email a related manuscript file with comments by our data editors. Please address all comments and upload a revised file with tracked changes with your final manuscript submission. In case quantification is based on less than 3 independent experiments, please display the data as scatter blots instead of bar graphs. The use of statistical tests needs to be justified in such a case.
- Finally, EMBO reports papers are accompanied online by A) a short (1-2 sentences) summary of the findings and their significance, B) 2-3 bullet points highlighting key results and C) a synopsis image that is 550x200-400 pixels large (width x height) in .png format. You can either show a model or key data in the synopsis image. Please note that the size is rather small and that text needs to be readable at the final size. Please send us this information along with the revised manuscript.

Kind regards,

Martina

Martina Rembold, PhD
Editor
EMBO reports

Referee #1:

Cell biology would greatly enhance this paper. The authors have not done additional work, which may be understandable in the present climate. If the editors feel that the current climate does not allow the proper experiments to be done, then the paper has been adequately revised.

Referee #2:

The revised version of the manuscript by Rennie et al. contains several new experiments and addresses all the points I raised in my review. It shows that the IUb-D2 complex has an intermediated binding affinity between the ID2 and ID2Ub complexes indicating some structural rearrangement. Furthermore, it provides more detail regarding the FANCI R1285Q mutant and how this could be defective in FANCI and FANCD2 ubiquitination. Finally, elegant data is provided that explain the differences in Kd's measured by the EMSA and PIFE assays. Kd's are now only provided based on the PIFE assay which makes the data easier to understand.

I have no further reservations and recommend publication of this nice piece of work in EMBO reports.

Prof. Helen Walden
University of Glasgow
RI Mol, Cell, Systems Biol
University avenue
Glasgow G12 8QQ
United Kingdom

Dear Helen,

I am very pleased to accept your manuscript for publication in the next available issue of EMBO reports. Thank you for your contribution to our journal.

At the end of this email I include important information about how to proceed. Please ensure that you take the time to read the information and complete and return the necessary forms to allow us to publish your manuscript as quickly as possible.

As part of the EMBO publication's Transparent Editorial Process, EMBO reports publishes online a Review Process File to accompany accepted manuscripts. As you are aware, this File will be published in conjunction with your paper and will include the referee reports, your point-by-point response and all pertinent correspondence relating to the manuscript.

If you do NOT want this File to be published, please inform the editorial office within 2 days, if you have not done so already, otherwise the File will be published by default [contact: emboreports@embo.org]. If you do opt out, the Review Process File link will point to the following statement: "No Review Process File is available with this article, as the authors have chosen not to make the review process public in this case."

Should you be planning a Press Release on your article, please get in contact with emboreports@wiley.com as early as possible, in order to coordinate publication and release dates.

Please note that under the DEAL agreement of Jisc Member institutions with our publisher Wiley your paper might be eligible for open access publication in a way that is free of charge for the authors. Please contact either the administration at your institution or our publishers at Wiley (emboreports@wiley.com) for further questions.

See also <https://authorservices.wiley.com/author-resources/Journal-Authors/open-access/affiliation-policies-payments/jisc-agreement.html>

Thank you again for your contribution to EMBO reports and congratulations on a successful publication. Please consider us again in the future for your most exciting work.

Kind regards,
Martina

Martina Rembold, PhD
Editor
EMBO reports

THINGS TO DO NOW:

You will receive proofs by e-mail approximately 2-3 weeks after all relevant files have been sent to our Production Office; you should return your corrections within 2 days of receiving the proofs.

Please inform us if there is likely to be any difficulty in reaching you at the above address at that time. Failure to meet our deadlines may result in a delay of publication, or publication without your corrections.

All further communications concerning your paper should quote reference number EMBOR-2020-50133V3 and be addressed to emboreports@wiley.com.

Should you be planning a Press Release on your article, please get in contact with emboreports@wiley.com as early as possible, in order to coordinate publication and release dates.

Corresponding Author Name: Helen Walden

Manuscript Number: EMBOR-2020-50133V1